



# Wave climate in the Arctic 1992-2014: seasonality and trends

Justin E. Stopa[1], Fabrice Ardhuin[1], and Fanny Girard-Ardhuin[1]

[1]Univ. Brest, CNRS, IRD, Ifremer, Laboratoire d'Océanographie Physique et Spatiale (LOPS), IUEM, 29280, Brest, France

*Correspondence to:* Justin E. Stopa (justin.stopa@ifremer.fr)

**Abstract.** Over the past decade, the diminishing Arctic sea ice has impacted the wave field which is principally dependent on the ice-free area and wind. This study characterizes the wave climate in the Arctic using detailed sea state information from a wave hindcast and merged altimeter dataset spanning 1992-2014. The wave model uses winds from the Climate Forecast System Reanalysis and ice concentrations derived from satellites as input. The ice concentrations have a grid spacing of 12.5 km, which is sufficiently able to resolve important features in the marginal ice zone. The model performs well, verified by the altimeters and is relatively consistent for climate studies. The wave seasonality and extremes are linked to the ice coverage, wind strength, and wind direction. This creates distinct features in the wind-seas and swells. The increase in wave heights is caused by the loss of sea ice and not the wind verified by the altimeters and model. However, trends are convoluted by inter-annual climate oscillations like the North Atlantic Oscillation (NAO) and Pacific Decadal Oscillation. The Nordic-Greenland Sea is the only region with negative trends in wind speed and wave height and is related to the NAO. Swells are becoming more prevalent and wind-sea steepness is declining which make the impact on sea ice uncertain. It is inconclusive how important wave-ice processes are within the climate system, but selected events suggest the importance of waves within the marginal ice zone.

## 1 Introduction

Sea ice moderates the climate; therefore, the declining ice in Arctic Ocean has direct impacts on the Earth's albedo, meridional ocean circulation, biologic ecosystems, and human activities. Satellite measurements from the last 30 years reveal the ice has decreased by 0.45 to 0.51 million km$^2$ or -10.2 to -11.4 % per decade (Hartmann et al., 2013; Comiso et al., 2008). This has a dramatic and direct impact on the sea state because there is a larger expanse of ocean available for wave development (Thomson and Rogers, 2014). Ocean waves have important implications in the Arctic because they drive the upper ocean dynamics influencing the rich biological cycle (Tremblay et al., 2008; Popova et al., 2010). Near the Alaska coastline waves are causing erosion (Overeem et al., 2011), and as the ocean opens connecting the Atlantic and Pacific for transportation and commerce, sea state knowledge becomes increasingly important (Stephenson et al., 2011; Jeffries et al., 2013). Therefore it is crucial to improve our knowledge of waves in the Arctic Ocean.

Ice and wave interaction is a highly coupled two-way system. On the one hand, sea ice defines the shape and size of the basin controlling the available sea surface open to the atmosphere for wave growth; and on the other hand, waves break-up ice (Marko, 2003). The warming in the past decade has created declining sea ice cover (Zhang, 2005; Steele et al., 2008; Screen



and Simmonds, 2010; Cavalieri and Parkinson, 2012; Frey et al., 2015). A model simulation and data from altimeters of Wang
et al. (2015) and Francis et al. (2011) respectively show that with the increased open water in the Beaufort and Chukchi Seas,
wave heights are increasing. The objective of this manuscript is to describe the wave climate in the Arctic Ocean pole ward of
66°N with emphasis on resolving wave and ice features at appropriate spatial scales. This provides an opportunity to describe
the Arctic as a complete system and enables our results to be related to existing regional studies in the Nordic Seas (Semedo
et al., 2014), the Nordic and Barents Seas (Reistad et al., 2011), and Beaufort-Chukchi Sea (Francis et al., 2011; Wang et al.,

7 2015).

Analysis of historical wave observations including in-situ buoy measurements (e.g., Gemmrich et al., 2011), remotely sensed
waves from altimeters (e.g., Zieger et al., 2009), observations from voluntary observing ships (e.g., Gulev and Grigorieva,
2006), and microseisms (e.g., Husson et al., 2012) give us rich information about the wave climate. Still, in the Arctic these
sources are not entirely satisfactory; therefore, we can use numerical models to provide full space-time details essential for
a comprehensive description of the wave conditions. WAVEWATCH III (called WW3 herein) of Tolman et al. (2013) is a
community based spectral wave model that describes wave evolution at the desired spatial and temporal resolution and provides
the full sea state details through a frequency-direction spectrum. In recent years there has been significant improvement to WW3
through the National Partnership Program (NOPP) which developed the physical parameterizations important for deep water
wave evolution (Tolman and the WAVEWATCH III Development Group, 2014).
To properly hindcast waves in the Arctic using WW3, it is imperative to use high quality sea ice information and wind in
the model. Ice concentrations from derived from the Special Sensor Microwave Imager (SSM/I) sufficiently resolve important
features in the marginal ice zone (MIZ) and have more than two decades of observations. The continuous availability and
amount of observations make wind reanalysis data the most comprehensive for wave hindcasting. Wave hindcasting using wind
reanalysis datasets has a long history of successful applications including the National Center for Environmental Prediction
(NCEP) Reanalysis I (R1) (Wang and Swail, 2001), the European Centre for Medium-Range Weather Forecasts (ECMWF)
reanalysis ERA-40 (Uppala et al., 2005) and ERA-Interim (ERAI) (Dee et al., 2011),and the National Center for Environmental
Prediction (NCEP) Climate Forecast System reanalysis (CFSR) (Chawla et al., 2013; Rascle and Ardhuin, 2013). To efficiently
hindcast in the Arctic basin we implement WW3 on a curvilinear grid matching the spatial resolution of ice concentrations at
12.5 km (Rogers and Orzech, 2013).
We hindcast the wave conditions from 1992 to 2014 because this period coincides with available ice concentrations from
SSM/I and significant wave heights from altimeters starting in 1991. We provide more background information regarding the
model setup, input wind and ice fields, altimeter wave data, and our analysis methodology in section 2. Section 3 focuses
on validating our wave modeling efforts using wave heights from altimeters. We then describe the wave climate in section 4,
illustrating the seasonality, extreme conditions, and trends of the wave field over the last 23 years. In section 5 we demonstrate
the importance of wave-ice interaction through selected wave events. Finally, we discuss the results and give our conclusions
in sections 6 and 7 respectively.



## 2  Datasets, Model Implementation and Methodology

The Arctic Ocean is smaller in scale compared to other basins and is 7000 km at its widest point. The ocean is surrounded by a continental shelf with depths of 300 m. The center of the basin near the North Pole has depths greater than 4000 km and is ice covered. The combination of ice coverage and geography creates the different regional seas shown in Figure 1. The seven sub-regions: 1) Nordic-Greenland Sea, 2) Barents Sea, 3) Kara Sea, 4) Laptev Sea, 5) East Siberia Sea, 6) Beaufort-Chukchi Sea, and 7) Baffin Bay are used to generalize the climate. The following subsections describe the input and validation datasets, model setup, and analysis techniques.

### 2.1  Ice concentration from IFREMER/CERSAT (SSM/I)

Satellite derived ice concentrations are an invaluable data source to observe ice behavior (e.g., Frey et al., 2015). The Special Sensor Microwave Imager (SSM/I) brightness temperatures accurately estimate sea ice concentration (e.g., Liu and Cavalieri, 1998). The ASI algorithm of Kaleschke et al. (2001) uses a transfer equation that relates the polarization difference to ice concentration as a percentage. High frequency channels of SSM/I are used to estimate a daily average on a 12.5 km grid at IFREMER/CERSAT. This resolution enables important spatial features of the MIZ to be resolved (Ezraty et al., 2007). Figure 2 shows the minimum ice extent and total ice area for the period 1992-2014. The time series in the left panel confirms the continual decrease in ice coverage. The minimum sea ice coverage is occurring later in September from 1992-2014 with some inter-annual variability and/or anomalous years of 1997 and 2006. The period 1992-2002 is stable followed by an accelerated loss in recent years with the lowest on record occurring in 2012. The right panel shows the spatial view of the ice edge minimum from the years 1992-2002, 2002, 2007, and 2012. The East Siberia, Chukchi, and Beaufort Seas have the largest changes in ice cover making them more vulnerable to increasing waves.

### 2.2  Reanalysis wind fields

The NCEP CFSR and the ECMWF ERAI winds have been used to drive wave models, and produce high quality wave hindcasts (e.g., Chawla et al., 2013; Rascle and Ardhuin, 2013; Stopa and Cheung, 2014; Dee et al., 2011). The important advancements of CFSR with respect to its predecessors Reanalysis I and II consist of coupling between the ocean, atmosphere, land surface, and sea ice model, assimilation of satellite radiances, and increased horizontal and vertical resolution in the atmospheric model (Saha et al., 2010, 2014). The atmospheric model has a resolution of approximately 0.3° (37 km) and assimilates data in three dimensions. CFSR is available hourly and winds at 10 m elevation (U10) are used to drive the wave model. The ECMWF ERAI performs well over its predecessor ERA-40 with improved spatial resolution of 0.7° at 6 hour intervals (Dee et al., 2011). The dataset is consistent in time and does not have the same discontinuous features as CFSR; but, it is not able to resolve the upper percentiles (Stopa and Cheung, 2014).

It is not evident whether CFSR or ERAI is better suited to drive a wave model in the Arctic. Therefore a concurrent hindcast from 2010-2014 is used to assess the wind forcing differences on the wave field. Appendix A gives a detailed description of the results and here we summarize. The largest differences are in the upper percentiles and ERAI significantly underestimates





the extreme wave heights. In short, the model errors mirror those of the global basin (Stopa and Cheung, 2014). Due to the
importance of resolving the extremes, we use CFSR to re-create the waves from 1992-2014.

### 2.3 Significant wave heights from altimeters

Wave data from altimeters are essential observations for validation and observation. Satellite altimeters provide an extensive
source of wave information and have observations every 5.5 km for the hindcast duration. Significant wave heights ($H_s$) are
measured from active microwave sensors typically in the Ku or Ka bands under all atmospheric conditions. Once the data
is quality controlled and sensor biases are removed, $H_s$ errors are comparable to buoy measurements (Zieger et al., 2009;
Sepulveda et al., 2015). We use the merged and calibrated dataset of Queffeulou and Croize-Fillon (2015). The reprocessed
wave measurements from European Remote Sensing Satellites 1 and 2 (ERS1,ERS2), Environmental Satellite (ENVISAT),
Geosat Follow-On, CRYOSAT2, and Altika SARAL are used throughout this study. The northern latitude limit is 81.4°N for
Geosat Follow-On, 82°N for ERS1, ERS2, ENVISAT, and SARAL, and 88°N for CRYOSAT2. The repeat track cycle is 17
days for Geosat Follow-On, 35 days for ERS1, ERS2, ENVISAT, and SARAL, and 369 days for CRYOSAT2.

### 2.4 WAVEWATCH III Model Implementation and Wave-ice Dissipation

Wave data is generated using the spectral wave model, WW3 version 5.08. WW3 evolves the wave action equation in space
and time, with discretized wave numbers and directions. Conservative wave processes, represented by the local rate of change
and spatial and spectral transport terms are balanced by the non-conservative sources and sinks. A curvilinear grid is well
suited to model waves near the Poles since the geographic distance between nodes is equal making the computation more
efficient (Rogers and Orzech, 2013). We use WW3's third order the Ultimate Quickest scheme of Tolman (2002) with the
garden sprinkler correction. Global 0.5° resolution hindcast of Rascle and Ardhuin (2013) provides the spectral boundary
conditions along 66°N. The spatial grid matches the ice resolution of 12.5 km and the spectra are composed of 24 directions
and 32 frequencies exponentially spaced from 0.037 to 0.7 Hz at a relative increment of 1.1. The source terms of Ardhuin et al.
(2010) describes the wave physics which performs well in terms of $H_s$, average wave periods, and partitioned wave quantities
(Stopa et al., 2015). The wind-wave growth parameter $\beta_{max}$ is set to 1.25 and otherwise we use the same settings as Rascle and
Ardhuin (2013). Due to the importance of wave-ice interactions, a new source term is developed and implemented in WW3 to
describe wave dissipation under ice. The dissipation is modeled by a laminar to turbulent boundary layer based on a critical
Reynolds number computed from the orbital wave velocity. This parameterization was calibrated using Wadhams and Doble
(2009) dataset but still remains somewhat poorly constrained. A full description of the formulation is given in Appendix B.

### 2.5 Wave parameters and analysis techniques

The wave climate is described using the total significant wave height ($H_s$) defined as $H_s = 4\sqrt{m0}$ where $m0$ is the zeroth
moment of the spectrum, average wave period ($Tm02 = \sqrt{m0/m2}$), and average direction ($\theta_m$). Swells characterized by
longer wavelengths propagate considerable distances under sea ice while high frequency waves are scattered and dissipated





near the ice edge (Kohout et al., 2014; Li et al., 2015; Ardhuin et al., 2015). Therefore the wind-waves ($H_{sw}$, $\theta_{\mathrm{mw}}$) and swells
($H_{ss}$, $\theta_{\mathrm{ms}}$) are analyzed separately by partitioning wave spectra using the Hanson and Phillips (2001) method. According to
Pierson and Moskowitz (1964) the sea state can be classified as wind sea when the wave age (WA) or ratio of peak phase speed
$C_p$ to wind speed, WA=$C_p/U10 < 1.2$ and swell when WA $>1.2$. Semedo et al. (2011) and Semedo et al. (2014) demonstrated
the practicality of this classification through the probability of having a swell dominated wave field (swell persistence): $P_s =$
$P(C_p/U10 > 1.2) = N_s/N_{total}$ where $N_s$ is the number of swell dominated events and $N_{total}$ is the total number of events.

7       In seas with varying ice cover, the method to describe wave statistics becomes increasingly important (Tuomi et al., 2011).

We base our statistics on ice free conditions (ice concentration <15%), but other statistics can be inter-related through the
sea ice probability shown in Appendix C. Our results are based on the 3-hour model output for the hindcast duration. $H_s$
percentiles are calculated from the ice-free statistics and the matching $H_s$ index is used to identify corresponding wave periods
and directions. A $\pm 0.2$ m bounds of the associated $H_s$ index is used to average the wave periods and directions. This approach
gives a more accurate physical description of the events (Anderson et al., 2015). We compute the rate of change using Sen's
slope and test for significance using the seasonal Mann-Kendall test. This method is a non parametric technique and a robust
way of computing trends since it can handle missing data and is less influenced by outliers (Mann, 1945; Kendall, 1975;
Sen, 1968). The method is generalized to account for the seasonal cycle by Hirsch et al. (1982) and has been used in wave
climatology studies by Wang and Swail (2001), Young et al. (2011), and Stopa and Cheung (2014). We compute trends from
monthly statistics and require that the time series must be ice-free for at least 10 years to reduce spurious features.

## 3   Wave hindcast validation

Before the wave climate is assessed we validate the model using the merged altimeter dataset for 1992-2014. Altimeter-model
co-locations are found using the nearest neighbor within 6 km and 30 minutes. A running mean of 5 points smooths the satellite
tracks to make the spatial and temporal scales comparable. The top 4 panels of Figure 3 show four complementary $H_s$ statistics
computed in 25 km bins. There is minimal bias for the majority of the domain; however, some errors exist. In the Baffin Bay
and the region north of Svalbard there is an underestimation of 5% and 10% respectively. Otherwise spurious negative biases
are located near the coasts. In the Beaufort, Chukchi and Laptev Seas the model overestimates the wave heights by 5-10%.
The RMSEs are commonly less than 0.4 m. East of Greenland has the largest RMSE of 0.5 m. This area has considerable
ice displacements within one day and the SSM/I daily input might not be able to resolve the rapid change. The Nordic Seas,
which are ice free year round, have the lowest scatter indices (12%) suggesting the model performs well far from the ice. In
contrast regions with considerable ice variability have the largest scatter indices of 30%. The model and altimeters are highly
correlated with coefficients larger than 0.95. The lowest correlations occur in the Laptev, East Siberia, and East Beaufort Seas.
These regions have small wave heights and result in small biases and RMSEs.
The bottom panel of Figure 3 shows the consistency of hindcast using the monthly 95th percentile for each satellite mission.
The 95th percentile is a rigorous test because it is difficult to resolve extreme waves. The average and median are verified
to have no distinct trends (not shown). There is a slight decreasing trend in recent years but it is inconclusive if this trend





will continue and is contrary to the global increase observed by Rascle and Ardhuin (2013) for 2006-2011. There is annual
variations but the hindcast is relatively consistent in time making it applicable for climate studies. No noticeable discontinuities
exist and the hindcast can be used to adequately describe the wave climate in the Arctic Ocean.

## 4 Wave Climate

### 4.1 Seasonality

The Arctic Ocean experiences a dramatic seasonal cycle. Daily spatial averages of the seven regions in Figure 1 generalize
the sea ice, wind speed, and wave seasonality in Figure 4. Two different classifications are identified by the relative change in
seasonal ice coverage. The first group has small seasonal variation like the Nordic, Greenland, and Barents Seas while large
seasonality occurs in the Baffin Bay, Beaufort-Chukchi, East Siberia, Laptev, and Kara Seas. All regions have an ice minimum
in early September. The wind forcing follows a sinusoid for all seas except the East Siberia Sea with a maximum in January
and minimum in July. In regions with reduced ice cover, like the Nordic-Greenland and Barents Sea, the waves mirror the wind.
Wave heights in the Baffin Bay and Kara Sea, which remain 10% ice free from November to July, follow the winds and can
be described by a sinusoid. The wave height seasonal cycle in the Beaufort-Chukchi, East Siberia, and Laptev Seas is skewed
with an annual maximum in October. The antisymmetric pattern is created by the increasing wind speeds in September and
October and partial exposure to the atmosphere. Due to the relative difference between the ice and wind we define the winter
season as January-February-March (JFM) and the summer season as August-September-October (ASO).
Figure 5 presents the wave conditions for the two extremes: JFM and ASO. The sea state is described through the $H_s$,
$H_{sw}$, $H_{ss}$, and $P_s$. In JFM, only the Nordic, Greenland, and Barents Seas are ice-free. Waves generated in the North Atlantic
propagate into these seas with a sheltering in the Barents Sea. The Nordic-Greenland Sea has the tallest wave heights of the
Arctic this time of year. The wind-waves follow the driving winds characterized by cyclonic (anti-clockwise) circulation, which
is a characteristic of the North Atlantic sub-basin (Sterl and Caires, 2005; Semedo et al., 2014). The resulting waves exceed 3.5
m while swells have smaller heights and travel from the Southwest. Even waves generated in the basin are classified as swells
70% of the time. This is consistent with open ocean conditions where there are always wind-waves and swells (Chen et al.,

24 2002).

In ASO, ice coverage is minimum and waves are generated across the Arctic. The $H_s$ pattern in the Nordic, Greenland, and
Barents Seas is similar to JFM with only a reduction in magnitude. The other seas have smaller $H_s$ than the Nordic-Greenland
Sea and are commonly less than 1.5 m. There are distinct regional characteristics of the wind-seas and swells. The cyclonic
structure in the wind-seas near Norway in JFM is not clearly visible in ASO. The swell directions follow the same pattern
as JFM in the Greenland, Nordic, and Barents Seas and flow from the Atlantic northward into the sea ice. In the Laptev and
East Siberia Seas the wind-waves and swells are directed into the ice with an Easterly component. $H_{sw}$ and $H_{ss}$ local maxima
located near (170°E, 77°N) where the Easterly waves are able to sufficiently develop. In the Beaufort Sea the wind-sea and
swell directions flow from the Southeast. In the East Beaufort Sea (135°E, 74°N), there is a subtle anti-cyclonic (clockwise)
structure in the wind-seas while the swell directions are opposite and flow from the West. In the narrow corridor of the Baffin





Bay, the wind-sea and swell directions are opposite and represent different storm phases. The Greenland, Nordic, Barents Seas,
and Baffin Bay are exposed to swells generated in the North Atlantic and thus dominated by swells >85% and exceed 95%. In
the semi-enclosed seas like the Kara, Laptev, East Siberia, Beaufort, and Chukchi there is an equal proportion of wind-waves
and swells (40-60%).

## 4.2 Percentiles

$H_s$ percentiles are a useful to way to describe the sea state statistical distribution (e.g., Stopa et al., 2013a, b). Figure 6 shows
the 50th (median), 95th, and 99th $H_s$ percentiles with matching wave directions and average periods. The statistics have a
consistent spatial pattern due to the geographic shape of the basin. The most prominent feature is the maximum located in the
Nordic-Greenland Sea for all percentiles. Here the median $H_s$ exceeds 2.5 m and corresponding $Tm02$ is 6 s. In the rest of
the basin the $H_s$ median is commonly 1.5 m with reduction near the coasts. The 95th $H_s$ percentile exceeds 5.5 m with $Tm02$
of 8 s in the Nordic-Greenland Sea with a reduction in the Barents Sea. In the Laptev, East Siberia, Chukchi, and Beaufort
Seas the 95th $H_s$ percentiles are 2.5 m with $Tm02$ of 6 s. The $H_s$ and $Tm02$ at the 99th percentile exceed 8 m and 9 s in the
Nordic-Greenland Sea while in the Laptev, East Siberia, Chukchi, and Beaufort Seas are reduced with 4 m and 6.5 s.
The wave directions give an indication of the physical nature of the events supporting the approach of finding the matching
indices of the $H_s$ percentiles. The directions of the median are less representative of physical processes because numerous
events affect this regime. In the Beaufort-Chukchi Sea the wave directions of the 50th percentile are focused into the ice pack
while for the 95th and 99th percentiles the wave directions flow from the East parallel to the ice edge. Due to the geometry of
the Beaufort-Chukchi Sea, the largest fetch occurs when the wind is parallel to the ice edge. So it is not surprising the extreme
waves are directed from the East. The wave directions of the 95th and 99th percentiles contain similar patterns as the seasonal
components of Figure 5. For example the region near East-Greenland is characterized by waves from the North as seen in the
wind waves while the waves offshore of Norway are directed from the South typical of swells in JFM. In the Beaufort, Chukchi,
and East Siberia Seas the East waves are common to both the wind-waves and swells in JFM and ASO. The percentiles of the
semi-enclosed seas have similar magnitude as the Gulf of Mexico (Stopa et al., 2013a).

## 4.3 Trends

With the loss of sea ice it is expected that wave heights are increasing. Figure 7 shows Sen's slope with the seasonal Mann-
Kendal test for ice coverage and $H_s$ from altimeters and the model. Trends are computed from the monthly averaged quantities
with removal of the seasonal cycle. The top left panel displays the trend of the SSM/I ice concentrations in number of days
per year that are ice-free (i.e. concentration <15%). Most of the ice covered areas are statistically significant and are ice-free
2 additional days each year. The strongest trends are located in the Barents and Kara Seas with 8 more ice-free days per year.
Only isolated regions near Svalbard, Greenland, and the Amundsen Gulf have increasing ice coverage.
Most of the basin has increasing wave heights shown by the altimeters and wave model in the top right and bottom panels.
The bottom panels show the co-located $H_s$ trends from the altimeters and the model agree, despite the stronger trends in
the altimeters. However, the altimeter confidence interval encompasses the model results so statistically they are equivalent.





Discrete satellite passes do not capture the complete space-time history causing spurious trends especially near the MIZ in the
East Siberia and Beaufort Seas. The trends computed from the continuous hindcast in the top right panel shows a spatially
consistent pattern. The ice variability is expected to cause the discrepancies in the East Siberia and Beaufort-Chukchi Seas that
exist comparing top panel bottom panels. The Nordic-Greenland Sea in the only region with a consistent statistically significant
decreasing trend shown in the top right panel. In the Beaufort-Chukchi Sea, our rates of 1.5 cm/year are in agreement with
Francis et al. (2011) who estimated a trend of 2 cm/year. Wang et al. (2015) estimated trends on the order of 40 cm computed
by the difference between 1970-1991 and 1992-2013. Assuming a linear rate spanning the 23 year period equates to a 35 cm
increase. Some extreme trends greater 4 cm/year exist in the Baffin Bay and Laptev Sea and are significant using the merged
altimeters. These rates are very large compared to the global calculations of Young et al. (2011), who estimated the largest
trends to be 2 cm/year.
Figure 8 shows the trends from other parameters using monthly averages. In this case the results are presented as percentages
relative to the mean to allow comparison. The trends in U10 are calculated using the entire dataset independent of ice cover;
otherwise all other variables are computed from ice-free statistics. The decreasing U10 trend in the Nordic Sea is significant
and is consistent with the $H_s$ trend. Across most of the sea ice, U10 is decreasing especially in the Beaufort Sea. Some regions
have weak increasing trends of 0.25% per year. Wind speeds in the Baffin Bay are increasing creating taller wave heights.
The trends in $Tm02$ follow the same pattern as the wave heights in Figure 7 and with an increase of 2% (2-3 cs) per year.
The trends in $H_{sw}$ and $H_{ss}$ heights have similar spatial patterns as $H_s$. However, $H_{ss}$ is increasing at a faster rate compared
to $H_{sw}$ in the Beaufort, East Siberia, Laptev, and Kara Seas. This is directly related to the higher occurrence of swells (i.e.
$WA$ is increasing). The decrease in $H_{ss}$ in the Nordic Sea has less statistically significant points suggesting changes in the
local winds are causing the trends. The bottom-center panel displays the wave steepness ($STw$) (ratio of wave height versus
wavelength) of the wind-sea components. The majority of the basin has a reduction in steepness illustrating the wavelengths
are increasing faster than the heights. The trends in swell steepness follow the same pattern as $H_s$ suggesting the wavelengths
are changing proportionally to the heights (not shown). Finally the $WA$ is increasing across the entire domain albeit some
decreasing regions near the MIZ in the Beaufort Sea, Greenland Sea, and Baffin Bay. Consequently the wave phase speeds are
increasing faster than the driving wind fields and swells are becoming more prevalent.
Trends often contain a component of variability which may lead to opposite trends in the future. The North Atlantic Oscil-
lation (NAO) has a strong influence in the Nordic-Greenland Sea shown by Semedo et al. (2014) and in the Beaufort-Chukchi
Sea, the Pacific Decadal Oscillation (PDO) influences the ice and wind dynamics (Frey et al., 2015). Table 1 presents corre-
lation coefficients between area-averaged monthly time series of sea ice, U10, and $H_s$ and the NAO and PDO indices. The
Southern Oscillation Index (SOI) generally had similar correlation coefficients as the PDO but where reduced in strength; so,
we only include the PDO here (see Appendix D for spatial distribution). The statistically significant relationship in the wind
and wave fields with the NAO is moderate in the the Nordic, Greenland, and Barents Seas. The PDO is weakly related in the
Beaufort-Chukchi Sea. Higher values are attained by correlating the time series for each point (see Appendix D).
Figure 9 graphically summarizes the regional trends through Sen's slope of the NAO, PDO, ice-free area, U10, and $H_s$.
The NAO and PDO have statistically significant decreasing trends. The NAO is expected to cause the decreasing trend in the





Nordic, Greenland, and Barents Seas seen in Figures 7 and 8. The most prominent feature is the increase in ocean area due to
the loss of sea ice. The $H_s$ trends are not homogeneous showing the regional variability. The largest trends in ocean-area and
$H_s$ occur in the Laptev and East Siberia Seas. All seas except the Baffin Bay have stronger trends in the average $H_s$ compared
to the 95th percentile suggesting non-uniform changes in the statistical distributions. When the average trend is higher than
the 95th percentile it means that events occur more frequent. In the Baffin Bay the trends in the 95th percentile are larger than
the average suggesting the intensification of strong events. The wave trends computed for the co-located altimeters and model
show the model underestimates (consistent with Figure 7). The $H_s$ average and 95th percentile often had larger trends than
the results in the global ocean which were typically less than 1% (Young et al., 2011). The trivial U10 trends illustrate the
increased sea states are due to ice loss in agreement with Wang et al. (2015).
## 5   Wave impact on the sea ice
So far, we have seen that the decreasing the sea ice has drastic impacts on the wave climate and has amplified sea states.
The waves also impact sea ice and their influence is unknown due to lack of observations and understanding of the wave-ice
processes. Large storms affect the Arctic as demonstrated by Simmonds and Rudeva (2012) and Zhang et al. (2013); however
the impacts from waves are less evident. Therefore we qualitatively describe the wave influence on the sea ice with selected
events. From the previous analysis, two distinctly different environments emerged. The first is in the Nordic-Greenland Sea
and is influenced by swells from the North Atlantic; and, the second is Beaufort-Chukchi Sea characterized by an equal mix of
wind-waves and swells coupled with an extreme change in seasonal ice coverage.
Figure 10 shows the Nordic-Greenland Sea area-averaged ice and sea state conditions for two months in 1992. The 23-
year average (dashed line) shows the ice cover climatology increases 6% from November through December . The first event,
November 23-29, indicates a decrease in ice cover by -3% corresponding to a loss of  60,000 km$^2$. This event coincides with
$H_s$, peak periods, and wind speeds exceeding 5 m, 12 s, and 14 m/s respectively. The second event in December has larger
wave heights (>6 m), however the ice cover remains the same.
We qualitatively compare and contrast these two events by considering the physical processes of the wind and wave condi-
tions. Figure 11 shows snapshots of the peak wind speed, wave height, and period for the two selected events. The white and
black lines denote the ice edge defined by 15% ice concentration before and after the event. During the November event, the
wind field rotation is cyclonic and centered near Iceland (14°W, 66°N). An anti-cyclonic pattern (6°E, 78°N) adds to the effec-
tive fetch. U10 exceeds 20 m/s and $H_s$ exceeds 9 m close proximity to the ice with wave periods ranging 12-15 s. Further into
the sea ice only the largest wave periods remain due to the attenuation of the short wavelengths. The wind and wave directions
are largely perpendicular to the Greenland ice edge. The largest sea ice changes are located from 70 to 77°N corresponding
to the maximum wave energy and wind speed. The bottom panels of Figure 11 show the December storm is located further
(7°E, 71°N) from the Greenland ice edge. This creates reduced wind speeds (<18 m/s), wave heights (3 m), and periods (12 s)
close proximity to the ice edge. There is minimal change to the ice edge during the event and is expected to be related to the





reduced amount of wind and wave energy entering the ice. We do not consider the ice thickness in our analysis; therefore it is
not apparent how much ice volume is lost by either event.
Figure 12 shows the ice cover and sea state conditions in the Beaufort-Chukchi Sea in September and October 2006. This
time of year the ice increases and advances southward. The September 13-16 and October 9-11 events both have sea ice losses.
In the first event the ice coverage reduces by 12% equating to 226,000 $km^2$ while in the second event the decrease of 6%
equates to 113,000 $km^2$. The second event has $H_s$ of 6 m, which is well above the climatology average of 2 m. The sea state is
much weaker in the first even than in the second.
Figure 13 illustrates the corresponding sea state conditions for the events. The September case has winds predominately
from the South directed into sea ice from the Bering Strait. The wind speeds are expected to be strengthened by the pressure
gradient force created by the tall mountain ranges in Alaska and Russia which exceed 2 km in height. The wind speeds, wave
heights, and periods reach 18 m/s, 4.5 m and 9 s offshore of the ice. The area of polynya located near (158°W, 77°N) does not
change much and is translated towards the Pole. There are significant changes to the ice edge and a large indentation coincides
with the maximum wind and wave energy. Besides this isolated section, the rest of the Beaufort ice edge remains the same.
The October event has large wind speeds, wave heights, and wave periods exceeding 24 m/s, 8 m, and 13 s. In fact, this
is the strongest event from 23-year hindcast and $H_s$ exceeding 8 m is well above the 99th percentile of Figure 6. The wind
field has a cyclonic pattern (low-pressure system) centered in the Chukchi Sea (162°W, 62°N) and an anti-cyclonic pattern
(high-pressure system) centered over the sea ice (132°W, 80°N). The positions of these systems create an extended fetch for
wave development because the Eastern winds are directed parallel to the sea ice. Ekman transport could be moving warmer
water towards the sea ice. In addition, the extreme wind and waves enhance mixing which could transport warm waters to the
ice. As time evolves the systems move further north creating a larger Southerly component in the Eastern portion of the domain
where significant impacts are made to the ice edge.
These examples suggest a relationship between the evolution of sea ice and the amount of wind and wave energy that is
directed into the MIZ. Therefore the orientation of the generating weather systems play a critical role on ice evolution. Other
physical processes that influence sea ice include temperature change, ocean circulation, and transport due to wind (Frey et al.,
2015). And these examples illustrate how waves are expected to impact the sea ice and should be considered as a potential sea
ice driver.

## 6   Discussion

Coupling between the waves and sea ice is complex (e.g., Squire et al., 1995; Squire, 2007). While the inclusion of the wave-ice
dissipation term is a step to incorporate improved wave-ice processes within the wave model, redistribution of the wave energy
through scattering must also be considered (Squire et al., 1995). Furthermore, wind-wave generation in partially ice covered
waters is expected to be more complex than present models (Li et al., 2015). Despite these missing physical processes, the
23-year hindcast presented here performs well offshore of the sea ice as demonstrated by the comparison to the altimeters. The





success of the hindcast is reliant on properly describing the sea ice. The SSM/I ice concentrations at 12.5 km resolution enables
our hindcast to capture important features in the MIZ.
The seasonal and extreme wave conditions in the Arctic are governed by the sea ice and winds. In most regions the wave
seasonal variation follows the winds and behaves like a sinusoid. In semi-enclosed regions like the East-Siberia, Beaufort, and
Chukchi Seas the seasonality is skewed with the maximum $H_s$ occurring in October which is due to concurrent increasing
winds and partially open seas. The isolation of these regions makes them event driven and they have an even mix of wind-
seas and swells. Extreme events are limited by the basin's size so the largest events have wave directions parallel to the ice
edge in order to attain the largest fetch. On the other side of basin, the Nordic-Greenland Sea has the most active sea states
within the Arctic because it is exposed to the Atlantic and is mostly ice-free year round. Therefore to a leading order the wave
behavior is linked to the geography and ice conditions which control the effective fetch for wave development. This supports
the non-dimensional fetch scaling of Thomson and Rogers (2014) to predict the wave conditions in the Beaufort Sea.
In the last 23 years, the loss of sea-ice but not the driving winds enabled wave heights to increase in our hindcast and
altimeter data sets agreeing with prior studies (Wang et al., 2015; Francis et al., 2011). The sea ice is becoming ice-free for
longer durations and the sea ice minimum is occurring later into September as Figure 2 suggests. If ice-free conditions persist
later into Fall, then seas with skewed seasonal cycles will be sensitive to this change because the wind is stronger this season.
Thus, the Kara, Laptev, East Siberia, Beaufort-Chukchi, and Baffin Bay will be prone to developing larger wave heights in the
Fall months when the wind speeds rapidly increase. This has lead Thomson and Rogers (2014) and Thomson and Team (2016)
to suggest a positive feedback mechanism linking enhanced wave heights to the larger ocean expanses which cause more ice
breakup. However this process is convoluted by the fact that the wave steepness is lessened which reduces the effectiveness of
the ice breakup by waves. So even though wave heights are increasing, the waves might not be as effective at breaking sea ice
because of their reduced steepness.
The wave response to the changing sea ice through the 21st century is complex with a mix of influences from wind, sea ice,
and climate variability (Khon et al., 2014). In our dataset, the natural variability of the climate through the NAO, PDO, and
SOI impact the Arctic. The NAO influences the Nordic-Greenland Sea and is expected to create the negative trend in U10 and
$H_s$ which is unique to the rest of the Arctic that has positive trends. The PDO influences the Barents and Kara Seas and the
monthly correlation closely aligns with the maximum ice loss trend. In the Beaufort-Chukchi Sea the PDO plays a minor role
in the wind and wave fields; therefore, the increasing waves are related to declining sea ice. When the PDO transitions into
a positive phase it might strengthen the Easterly winds, which flow parallel to the ice edge, in the Beaufort-Chukchi Sea and
result in larger sea states.
The impact of waves on the sea ice is difficult to determine without detailed knowledge of wave-ice interaction. The evolution
of the ice edge seems to response to the amount of wind and wave energy near the ice pack as demonstrated by computing
area-averages of wind and wave quantities in the Nordic-Greenland and Beaufort-Chukchi Seas. This supports the idea that the
orientation of the driving wind fields and their incident angles relative to the sea edge is important for sea ice evolution. For
example the Great Arctic Cyclone of 2012 persisted for 13 days and had a significant impact on the sea-ice (Simmonds and
Rudeva, 2012; Zhang et al., 2013). However, this storm did not produce substantial waves according to our hindcast, due to its





passage over ice-covered regions. If the location of the event was positioned more appropriately for wave development, then this type long-lived event could have produced large waves. Therefore if the duration of the storm events increases the sea states could achieve full development. The duration of the wind and wave influence is less evident in sea ice evolution. For example the percentage of ice cover remains relatively constant for a 10 day period after the strong wave event in the Beaufort-Chukchi Sea on October 9, 2006.

The sea ice variability is influenced by many drivers including atmospheric motion, oceanic motion, air/sea temperatures, and changes in cloud cover (Perovich, 2011). And waves should also be added to the list as important sea ice driver.

# 7 Conclusion

Extending previous studies in the Arctic we produced a 23-year wave hindcast from 1992 to 2014 using CFSR winds and ice concentrations from SSM/I. As the Arctic continues to evolve the results presented here can be used a basis for future climate conditions. Since the Arctic is semi-enclosed, the seas are event driven. The wave field is intertwined with the sea ice coverage and wind conditions where the wind direction defines the effect fetch for wave development. The loss of ice is creating enhanced sea states with influence from large scale climate oscillations. While it is not evident how important wave-ice processes are within the Earth-system, the increasing sea states in the Arctic have direct implications on other important processes. The increasing sea states are expected to change the air-sea fluxes. The reduced sea ice adjacent to land combined with the increasing sea states suggests wave driven coastal hazards and erosion will increase (e.g., Overeem et al., 2011). Waves induce turbulence and mixing in the upper ocean which are expected to influence the ecosystem (e.g., Tremblay et al., 2008; Popova et al., 2010). Furthermore higher sea states present larger risks to the growing number of ocean activities and ships operating in the Arctic.

The combination of wave models and satellite data provides sufficient detailed sea state information about the Arctic, and we are essential resources to monitor the changing climate.





**Appendix A:  The ECMWF Reanalysis Interim and the NCEP Climate Forecast System Reanalysis Arctic**
**Intercomparison 2010-2014**
Before the 23 year hindcast is implemented, two 5-year hindcasts using CFSR and ERAI are inter-compared in order to
establish a more suitable wave forcing. Observed wave data is essential for validation and we use buoys and altimeters. Only a
limited number of buoys are available from the National Data Buoy Network (NDBC) in the Chukchi Sea and their locations
are shown in Figure 1. These buoys are deployed and collected each season when the ice retreats providing data from July
through October each year. Only select years and $H_s$ measurements are available from 2012-to 2014. The buoys provide
essential ground truth for model validation, despite being located in depths less than 50 m with limited spatial coverage.
Table A1 displays error $H_s$ metric at the buoys. Standard error metrics including the normalized bias (NBIAS), root mean
square error (RMSE), correlation coefficient (R), scatter index (SI), and normalized standard deviation (NSTD) are given below
assuming the *x* represents the observation and *y* represents the model, and *n* is the number of data pairs:

$$NBIAS = \left[ (\overline{y} - \overline{x}) / \left( \frac{1}{n} \sum_{i=1}^{n} x_i^2 \right) \right] \times 100 \tag{A1}$$

$$RMSE = \sqrt{\frac{1}{n} \sum_{i=1}^{n} (y_i - x_i)^2} \tag{A2}$$

$$R = \sum_{i=1}^{n} (y_i - \overline{y})(x_i - \overline{x}) / \left[ \sqrt{\frac{1}{n} \sum_{i=1}^{n} (y_i - \overline{y})^2} \sqrt{\frac{1}{n} \sum_{i=1}^{n} (x_i - \overline{x})^2} \right] \tag{A3}$$

$$SI = \left[ \sqrt{\frac{1}{n} \sum_{i=1}^{n} [(y_i - \overline{y}) - (x_i - \overline{x})]^2} / \overline{x} \right] \times 100 \tag{A4}$$

$$NSTD = \left[ \sqrt{\frac{1}{n} \sum_{i=1}^{n} (y_i - \overline{y})} / \sqrt{\frac{1}{n} \sum_{i=1}^{n} (x_i - \overline{x})} - 1 \right] \times 100. \tag{A5}$$

All buoys have water depths less than 50 m. CFSR overestimates the $H_s$ by at least 5% at all locations while ERAI under-
estimates by 3% (except WMO48213). The RMSEs are commonly 0.25 m with ERAI always having a slightly better match.
The scatter indices and correlation coefficients for ERAI and CFSR follow the same pattern at each buoy. The NSTD shows
CFSR has more variability than the observations while ERAI is a smoother model. In general both models are comparable with
a positive bias in CFSR and negative bias in ERAI.
Figure A1 displays two example time series from September 2013 and 2014 at buoys WMO 48213 and 48214 located in the
Chukchi Sea. The first example in 2013 shows both models perform reasonably well. ERAI's time series is much smoother and





creates a correlation coefficient of 0.89. CFSR is seen to overestimate the events on September 1-3 and 26-28 with differences
larger than 0.5 m. For these events ERAI follows the same pattern suggesting a systematic error in the forcing wind field or
wave physics that are unresolved. The CFSR and ERAI residuals are moderately correlated with coefficients of 0.75 showing
the forcing wind fields are similar. In September 2014, both ERAI and CFSR are highly correlated to the buoy time series and
their residuals are only weakly correlated with a coefficient of 0.37. CFSR has a consistent positive bias throughout the month,
while ERAI commonly has errors less than 25 cm. The peak intensity of wave events is underestimated by ERAI.
The $H_s$ recorded from altimeters provides an extensive dataset for validation. We co-locate the $H_s$ from the altimeter and
model by finding the nearest neighbor in space (<6 km) and limit the time difference to 30 minutes. A running mean of 5
points is then used to smooth the altimeter tracks in order to make the spatial scales comparable between the computational
model grid and the altimeter. Figure A2 summarizes the results using the 5-year period of altimeter-model co-locations. The
scatterplots from all 2 million data pairs is presented in the top panels. Both datasets are highly correlated with similar SIs of
19% and have RMSEs of 0.4 m. The smooth nature of ERAI creates the negative NSTD of 10% while CFSR is nearly zero.
The largest differences in the hindcasts are in the upper wave heights and the bottom panels highlight the differences. Both
datasets have similar correlation coefficients of 0.78 and scatter indices of 13%. he CFSR has more variability than the observed
data creating a NSTD of 20% while ERAI matches the variability of the observations much better. From this depiction it is
clear that ERAI underestimates the largest wave heights. For example the 99th $H_s$ percentile has an average bias of -1.5 m
while its -0.1 m for CFSR. These large sea states are important to resolve in practical planning and engineering applications.
Therefore caution should be applied when using ERAI to describe the extreme waves. In our implementation of ERAI, $\beta_{max}$
described in Ardhuin et al. (2010) is set to 1.45 and possibly a better match could be achieved by increasing this value. An
empirical correction could also be applied similar to Sterl and Caires (2005) with ERA-40.
In the previous comparison it is clear that the upper percentiles are very different. Now, we compare the statistical distribu-
tions spatially. Figure A3 compare the median as well as the most extreme wave conditions at the 99th percentile. The top left
and center panels display the percentiles from CFSR and the differences between CFSR and ERAI are in the bottom panels.
The medians are clearly different and the ERAI is less than CFSR by 0.1 to 0.4 m across the Arctic. The extreme wave heights
have large differences of 2 m in the area East of Greenland that has significant wave heights larger than 7.5 m. Otherwise the
waves in ERAI are 0.5 m less than CFSR across the semi-enclosed seas. The top right panel shows the datasets are highly
correlated with coefficients larger than 0.95 and exceed 0.98. The Mann-Whitney test reveals that the medians come from
different statistical distributions at the 99.9% confidence limit for the entire domain. Therefore we can conclude that the largest
differences are in the extreme events but the medians are different as well.
Finally the probability distributions using the buoy and altimeter observations are presented in Figure A4. CFSR matches
the wave heights larger than 2.5 m well, while ERAI consistently underestimates. When the $H_s$ is less than 2.5 m CFSR
overestimates more often. ERAI lends to favor the small wave heights. The comparison to the buoys in the right panel shows
similar features and it is clear that CFSR overestimates average wave heights of 1-2.5 m which agrees with the examples shown
in Figure A1.



In conclusion, both datasets perform reasonably well and their results agree with errors found in the global ocean (Stopa and
Cheung, 2014). CFSR consistently predicts higher wave heights for average sea states and matches the upper percentiles much
better. Both ERAI and CFSR have better agreement with the measurements between the 10th and 99th percentiles. The upper
wave heights in ERAI diverge from the observations and the 99th percentile has an average bias of 1.5 m. In summary, ERAI
is better suited to describe the average conditions and the 6 hour increment and spatial resolution of 0.7 limits its ability to
resolve the peak intensity of the storms. Due to the importance of resolve the upper wave heights we choose CFSR to hindcast
the entire period from 1992-2014.
**Appendix B:  Theoretical formulation of friction under ice plates**
**B1    Extension of the theory by Liu et al.**
The representation of dissipative source terms in spectral wave models can generally be cast in a quasi-linear form (Komen
et al., 1994)
$$S(f,\theta) = \beta\sigma E(f,\theta),\qquad\qquad(B1)$$
where $E(f,\theta)$ is the frequency-direction spectrum of the surface elevation, $\sigma = 2\pi f$, and $\beta$ is a non-dimensional dissipation
coefficient that is negative when wave energy is actually dissipated. Previous treatments of the dissipation of wave energy due to
friction below an ice layer have been confined to a laminar viscous boundary layer and presented by Liu and Mollo-Christensen

16  (1988),

$$\beta_v = -k\sqrt{\nu\sigma/2}/(1 + kM),\qquad\qquad(B2)$$
in which $k$ and $\sigma$ are the wavenumber and radian frequency, related by a dispersion relation that can be affected by the ice,
and $M$ is the ice inertia effect related to the ice thickness multiplied by the ratio of ice to water density. In the present paper,
because we focus on the dominant long-period waves, for which the effect of the ice is less, we have used the ice-free dispersion
relation $\sigma^2 = gk\tanh(kD)$ in which $D$ is the water depth and $g$ the acceleration of gravity. For these long waves, the factor
$kM$ in eq. (B2) can be neglected.
For practical applications, the obtained dissipation coefficient $\beta$ was then scaled up to fit observed wave attenuations by
replacing the molecular viscosity at the freezing temperature of sea water, $\nu \simeq 1.83 \times 10^{-6}$ m$^2$/s by an eddy viscosity that was
proposed to be as large as 0.3 m$^2$/s (Liu et al., 1991). Such a change in viscosity only makes sense if the flow is turbulent.
Further, the functional form of the dissipation can be very different for laminar and turbulent frictions in an oscillatory flow
near a boundary, as observed by Jensen et al. (1989). In turbulent boundary layers, the energy dissipation coefficient typically
grow with the wave amplitude, leading to a dependence of $\beta$ on the wave amplitude.
We thus revisit this question and propose a parametrization for the laminar to turbulent transition of the boundary layer
below the ice. In turbulent conditions, an important parameter is the roughness length below the ice $z_0$. That roughness is
unfortunately not well known, with only a few measurements of current boundary layers (e.g. McPhee and Smith, 1976).





Because the roughness for the wave motion is probably different from the roughness for the currents, as it is well known for ocean bottom boundary layers (Grant and Madsen, 1979), we are left with the difficulty of defining the value of $z_0$. Given this roughness, the orbital velocity profile is expected to follow a Kelvin function (Grant and Madsen, 1979) with a dissipation source term that takes a form similar to that of bottom friction (e.g. Madsen et al., 1990; Ardhuin et al., 2003) or swell dissipation by friction at the air-sea interface,

$$\beta_t = -f_e u_{\mathrm{orb}}/g \tag{B3}$$

where the significant orbital amplitudes of the surface velocity is, for deep water waves,

$$u_{\mathrm{orb}} = 2\sqrt{\int_0^\infty (2\pi f)^2 E(f)df}. \tag{B4}$$

and $f_e$ is the same dissipation factor used for bottom friction, that is a function only of the ratio $a_{\mathrm{orb}}/z_0$ where $a_{\mathrm{orb}}$ is the significant orbital displacement at the sea surface, here for deep water waves $a_{orb} = H_s/2$.

From bottom and air-sea boundary layer studies, the transition from laminar to turbulent is expected to occur for at a threshold $\mathrm{Re}_c$ of the significant Reynolds number defined by

$$\mathrm{Re} = u_{\mathrm{orb}} u_{\mathrm{orb}}/\nu. \tag{B5}$$

We take the same critical value $\mathrm{Re}_c = 1.5 \times 10^5$ found for the bottom boundary layer by Jensen et al. (1989) and the air-sea boundary layer by Perignon et al. (2014). Because of the random nature of the waves, with Rayleigh-distributed wave heights, we expect a smooth transition of the average dissipation rate from viscous to turbulent. We found that the average dissipation caused by random wave heights that follow a Rayleigh distribution is well approximated by the following combined dissipation parameter

$$\beta_c = (1-w)c_v\beta_v + wc_t\beta_t \tag{B6}$$

in which $c_v$ and $c_t$ are empirical adjustment constants, expected to be close to 1, and the weight $w$ transitions smoothly with the value of Re over a range $\Delta_{Re} = 200000$,

$$w = 0.5\left[1 + \tanh\left((\mathrm{Re} - \mathrm{Re}_c)/\Delta_{\mathrm{Re}}\right)\right]. \tag{B7}$$

Figure B shows the expected decay distance as a function of frequency, due to molecular viscosity (blue) or a turbulent boundary layer with a roughness $z_0=0.1$ mm, for significant wave heights ranging from 0.5 to 5 m. In our applications we have chosen $z_0 = 1$ cm.

## B2 Empirical adjustment of the wave attenuation

Wadhams and Doble (2009) have reported measurements of waves with periods larger than 20 s far into the ice pack (the periods reported in the paper were erroneously reduced by a factor 1.5, personal communication of M. Doble, 2015). An event





with 20 s waves recorded 1400 km into the ice pack on February 13, 2007, had a maximum significant wave height of 3 cm.
For this small wave height the wave boundary layer is expected to be laminar. However, using the dissipation coefficient in eq.
(B6) produced maximum wave heights of 30 cm. Changing only the coefficient $c_v$, it was necessary to increase it from 1 to 8
to obtain a reasonable agreement with the data. We have thus used that value to obtain reasonably small wave heights across
the Arctic.
However, we note that $c_v = 8$ tends to overestimate the dissipation in the Southern ocean case discussed by Ardhuin et al.
(2015), for which $c_v \simeq 2$ is a better adjustment. Such differences could be partly caused by a more complex geometry of older
ice in the Arctic, but a four-fold increase of the area of the ice-water interface that could explain this difference is unlikely. It
thus appears that the attenuation in the Arctic may be dominated by other processes than under-ice friction, especially when
the ice is not broken. Different processes probably produce different distributions of wave heights in the ice. Given the weak
energy level back-scattered in the open waters, the details of the wave attenuation process are not likely to affect much our
analysis of wave climatology outside of the ice.

**Appendix C:  Percentage of ice-free time**

In the manuscript we present ice-free statistics. Sea ice is assumed when the concentration is larger than 15%. The presentation
of the statistics will vary based on the method used to consider the inclusion of ice conditions in the analysis. Each method
of computing wave climate statistics described by Tuomi et al. (2011) has its own advantages depending on the application.
However the statistics can be related through the percentage of ice-free time presented in Figure C1. The colorbar is displayed
in a logarithmic scale to highlight the details of the small ice percentages while including the regions rarely covered by ice.
The Nordic-Greenland Sea is ice-free and the area closest to the North Pole is ice-covered throughout the year. It is clear from
this depiction that the largest changes occur in the Beaufort-Chukchi Sea and are ice-free less than 15% of the year above the
latitude of 74°.

**Appendix D:  Relationship with the North Atlantic Oscillation and Pacific Decadal Oscillation**

The manuscript presents area-average correlations with the NAO and PDO to indicate the strength of the relationship. These
values can be less for individual time series of each point. Therefore we compute the correlation coefficients between the
monthly average Hs and the climate indices for all grid points in Figure D1. This is a more accurate portrait of the strength of
the relationship and gives the full spatially distribution. It is clear the NAO has the strongest signature in the Nordic-Greenland
Sea and extends into the Barents Sea. The maximum correlation coefficient is 0.48 which is larger than Table 1 which is 0.37.
Other regions have reduced values of correlations and are not spatially homogeneous. The PDO has been largely negative
for the past decade and is creating the negative correlation coefficients across the Arctic. It is interesting to see that the largest
relationship occurs in the Barents Sea (R=-0.46) which the area-average results are much less (R=0.1). Only a weak relationship
exists in the Beaufort-Chukchi Sea contrary to what Frey et al. (2015) showed for the ice and wind field.





1  *Acknowledgements.* This work was supported by LabexMER through grant ANR-10-LABX-19. The CFSR and ERAI reanalysis data is

2  publicly available from rda.ucar.edu and the IFREMER/CERSAT sea ice concentration is available from ftp.ifremer.fr/ifremer/cersat/.



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





**Table 1.** Correlation coefficients and trends for the various regions and parameters. The correlations coefficients are given between area-averaged monthly time series versus the North Atlantic Oscillation and the Pacific Decadal Oscillation in parentheses. Statistically significant results are given by the '*' when the p-value is less than 0.05.

| Region | Avr Ocean Area | $U10_{Avr}$ | $U10_{P95}$ | $H_{sAvr}$ | $H_{sP95}$ |
|---|---|---|---|---|---|
| Arctic Ocean | -0.19*(-0.25*) | +0.37*(-0.02 ) | +0.35*(-0.08 ) | +0.31*(-0.02 ) | +0.31*(-0.08 ) |
| Nordic-Greenland Sea | -0.10 (-0.27*) | +0.37*(-0.09 ) | +0.35*(-0.11 ) | +0.32*(-0.08 ) | +0.33*(-0.13*) |
| Barents Sea | -0.15*(-0.33*) | +0.29*(-0.10 ) | +0.29*(-0.12*) | +0.22*(-0.04 ) | +0.22*(-0.10 ) |
| Kara Sea | -0.15*(-0.16*) | +0.19*(-0.20*) | +0.16*(-0.20*) | +0.11 (-0.10 ) | +0.05 (-0.14*) |
| Laptev Sea | +0.15 (-0.27*) | -0.01 (-0.11 ) | -0.01 (-0.09 ) | -0.12 (+0.06 ) | -0.08 (+0.01 ) |
| E. Siberia Sea | +0.17 (-0.13 ) | +0.01 (-0.08 ) | +0.02 (-0.06 ) | -0.12 (+0.14 ) | -0.01 (+0.13 ) |
| Beaufort-Chukchi Sea | -0.03 (-0.24*) | +0.05 (-0.21*) | -0.00 (-0.21*) | +0.05 (-0.15 ) | +0.00 (-0.17*) |
| Baffin Bay | -0.12 (-0.19*) | -0.07 (-0.16*) | -0.09 (-0.15*) | +0.10 (-0.11 ) | +0.08 (-0.13 ) |





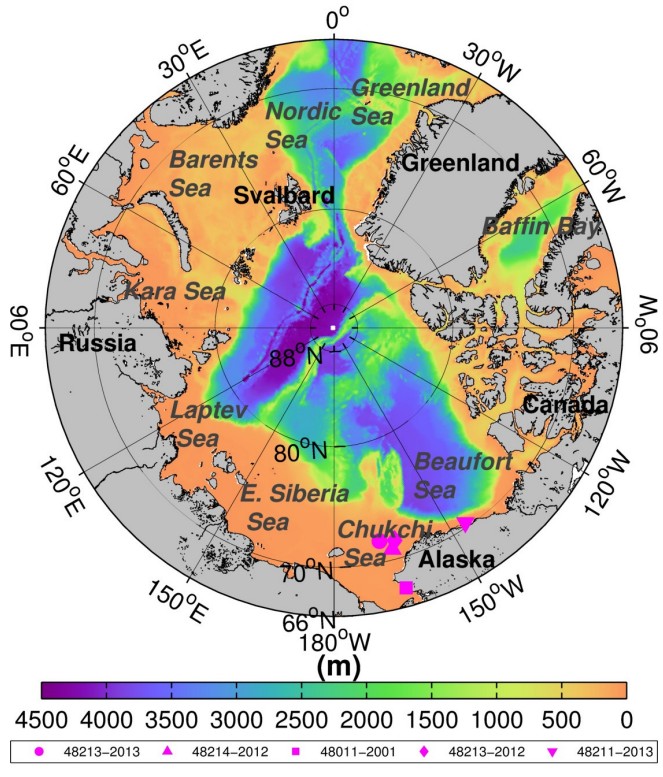

**Figure 1.** Regional seas of the Arctic Ocean colors respond to depth and buoy locations are plotted in magenta.





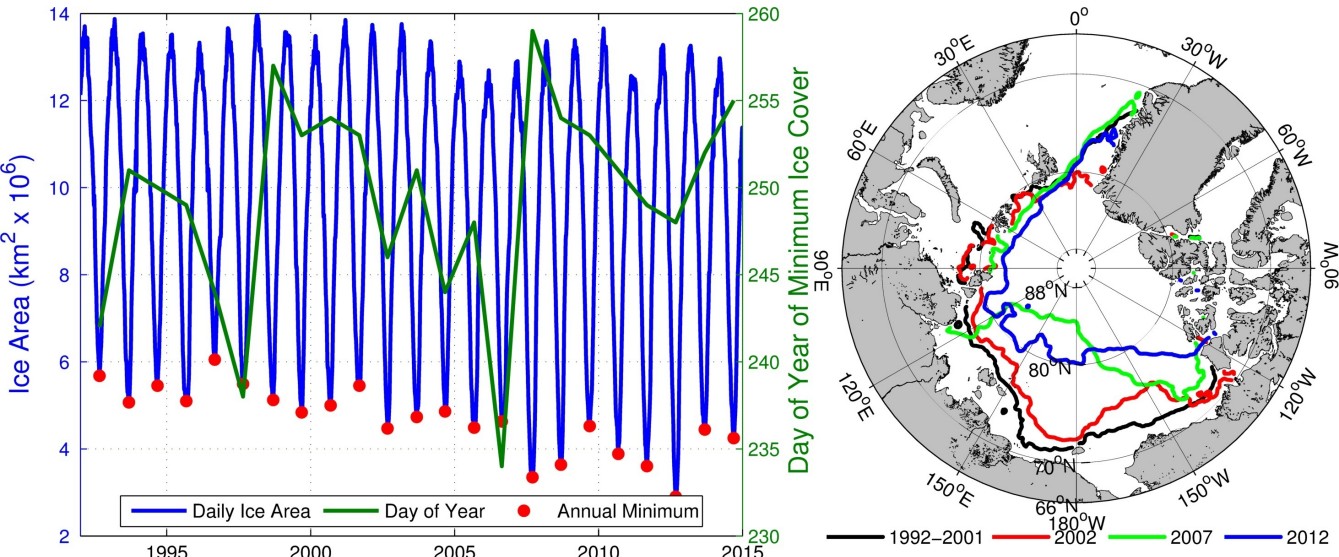

**Figure 2.** Daily ice area (left) and yearly ice area minimum for 1992-2001 (represents the median), 2002, 2007, and 2012 (right) from the Ifremer/CERSAT ice concentrations. The ice edge is defined as 15% ice concentration.





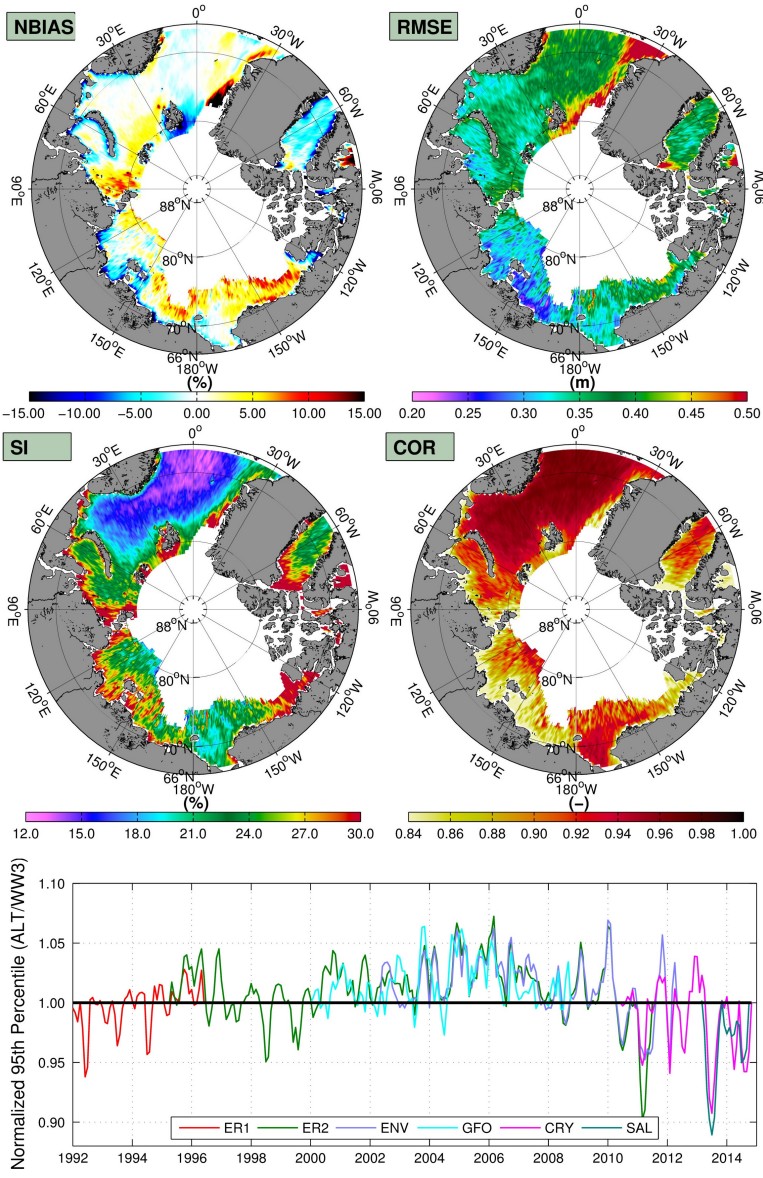

**Figure 3.** Normalized bias (NBIAS), root mean squared error (RMSE), scatter index (SI), and correlation coefficient (COR) for collocated significant wave heights for the CFSR wave hindcast and the merged altimeters 1992-2014 (top four panels). The bottom panel displays the normalized monthly $H_s$ 95th percentile for each satellite: European Remote Sensing Satellites 1 and 2 (ER1, ER2), Environmental Satellite ENVISAT (ENV), Geosat Follow-On (GFO), CRYOSAT2 (CRY), and Altika SARAL (SAL).





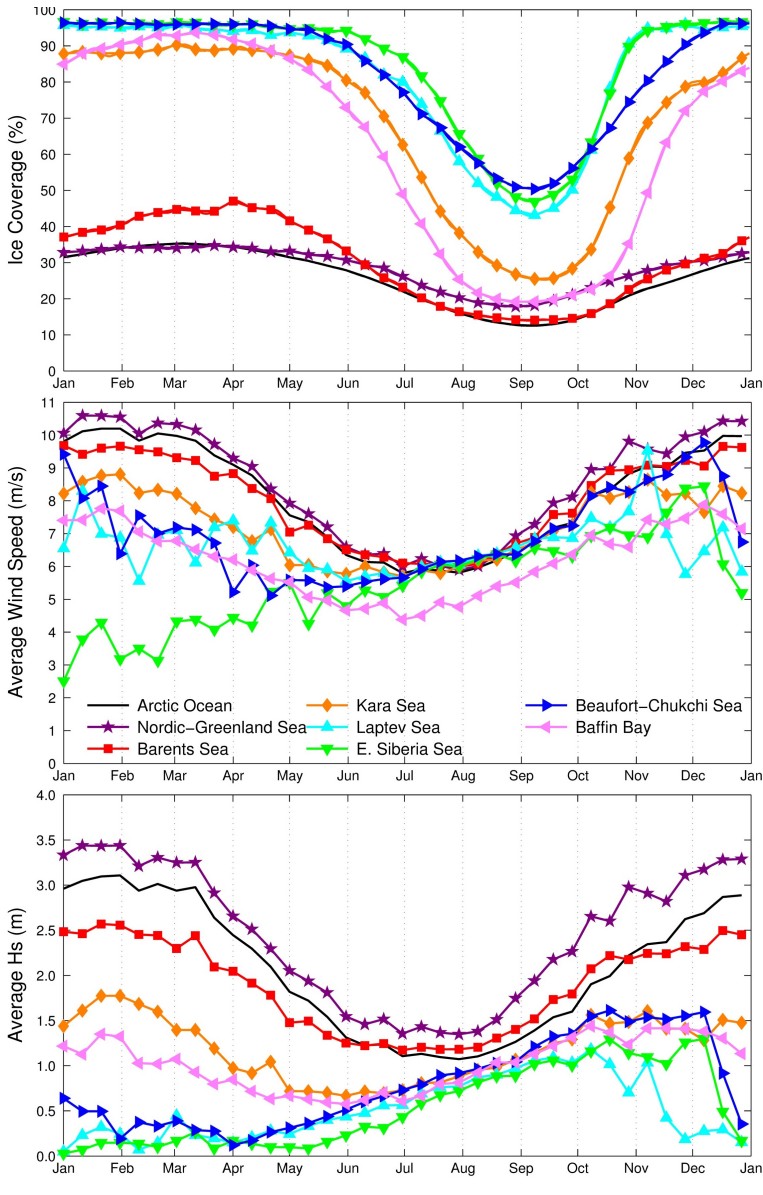

**Figure 4.** Daily averaged ice coverage, wind speed, and significant wave heights for different regions of the Arctic illustrating the seasonality.





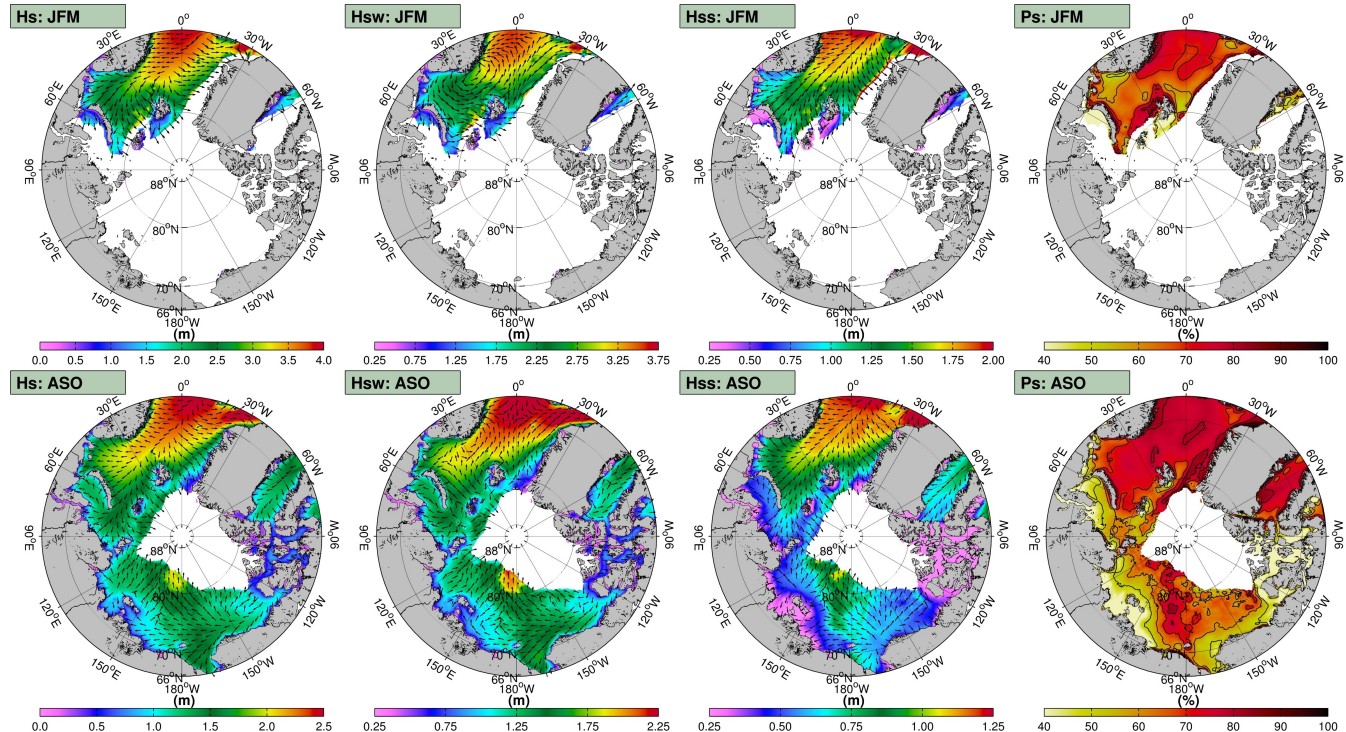

**Figure 5.** January-February-March (JFM) and August-September-October (ASO) seasonal averages of significant wave height $H_s$ (first column), wind sea wave height $H_{sw}$ (second column), swell wave height $H_{ss}$ (third column), and swell persistence $P_s$ (fourth column). The directions are computed from by averaging the east-west and north-south components separately.





**Figure 6.** Significant wave height percentiles (top panels) with matching wave periods (bottom panels) and directions.





**Figure 7.** Sen's slope with the Mann-Kendall test (thatched-areas) for yearly counts of ice free days per year (ICE) (upper left panel) and monthly averaged significant waves from the entire wave dataset (All) (upper right panel), altimeters (ALT) (lower left panel), and co-located wave model (lower right panel) given in terms of day cm per year.



**Figure 8.** Sen's slope with the Mann-Kendall test (thatched-areas) for monthly averaged wind speeds (U10) (top left panel), wind-sea wave heights ($H_{sw}$) (top middle panel), swell wave heights ($H_{ss}$) (top right panel), average wave period ($Tm02$) (bottom left panel), wind-sea steepness ($ST_w$) (bottom middle panel), and wave age ($WA$) (bottom right panel) from the wave model given in terms of percentage per year relative to the average.



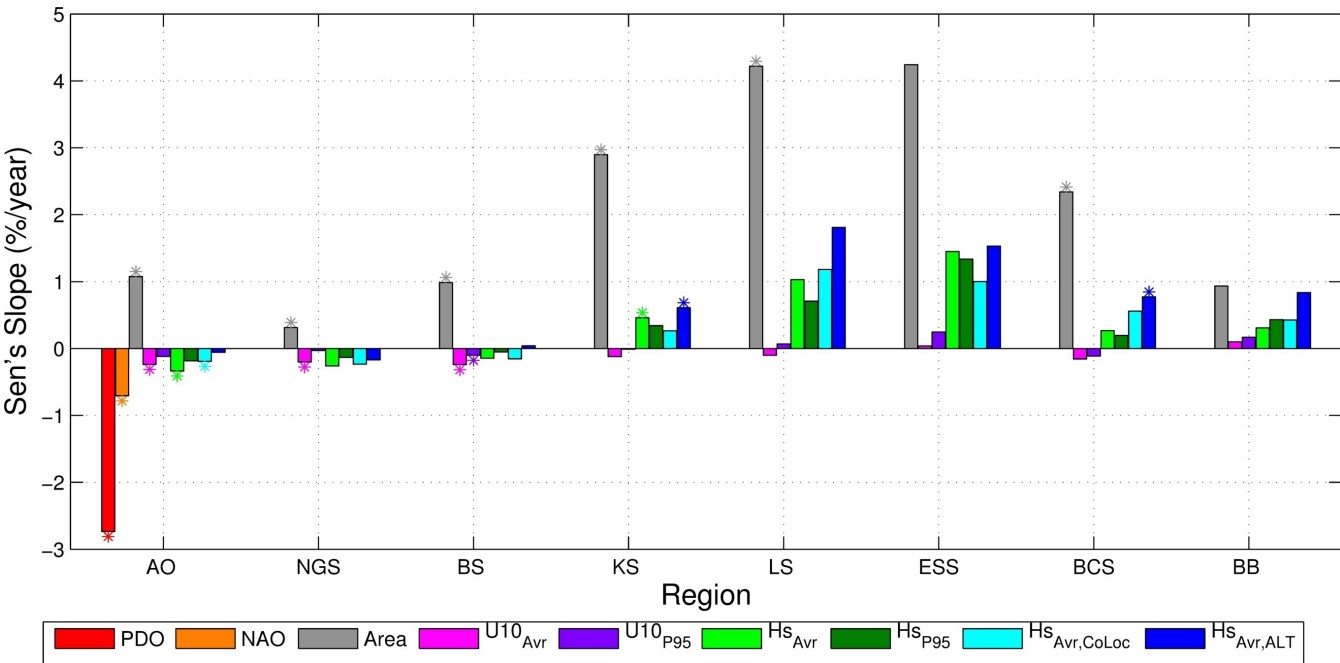

**Figure 9.** Sen's slope with the Mann-Kendall test (denoted by a '*') for the Arctic Regions of the Arctic Ocean (AO), Nordic-Greenland Sea (NGS), Barents Sea (BS), Kara Sea (KS), Laptev Sea (LS), East Siberia Sea (ESS), Beaufort-Chukchi Sea (BCS), and the Baffin Bay (BB) from monthly time series of the North Atlantic oscillation (NAO), Pacific Decadal Oscillation (PDO), ocean area (Area), wind speed (U10), significant wave heights using all model data ($H_{sAvr}$ and $H_{sP50}$), significant wave heights using co-located model and altimeter data ($H_{sAvr,CoLoc}$, $H_{sAvr,ALT}$).





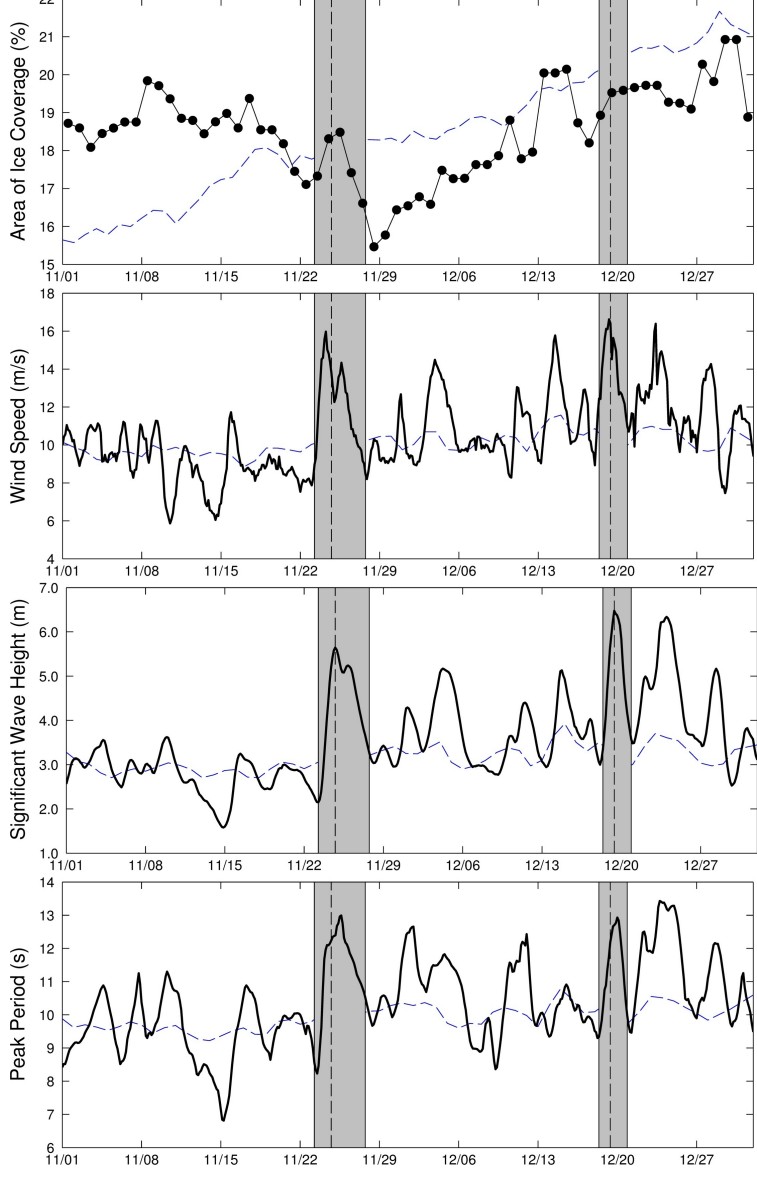

**Figure 10.** Area-averaged time series of ice coverage, wind speed, significant wave height, and peak period in the Nordic-Greenland Sea for November-December 1992 (solid line). The dashed line is the daily average from 1992-2014.





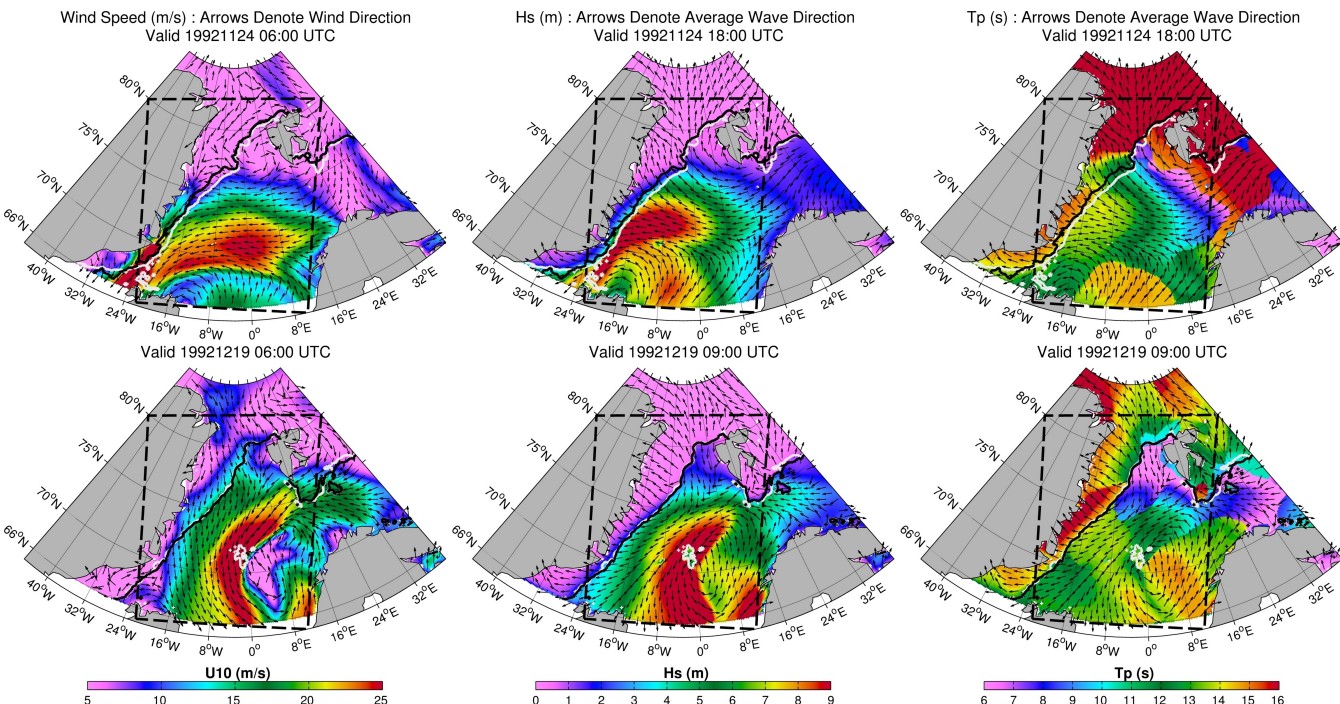

**Figure 11.** Wind speeds, significant wave heights, and peak periods for two selected events in November and December 1992. The arrows denote either the wind direction or average wave direction. The white and black contour lines represent the ice edge defined by a 15% concentration before and after the event.





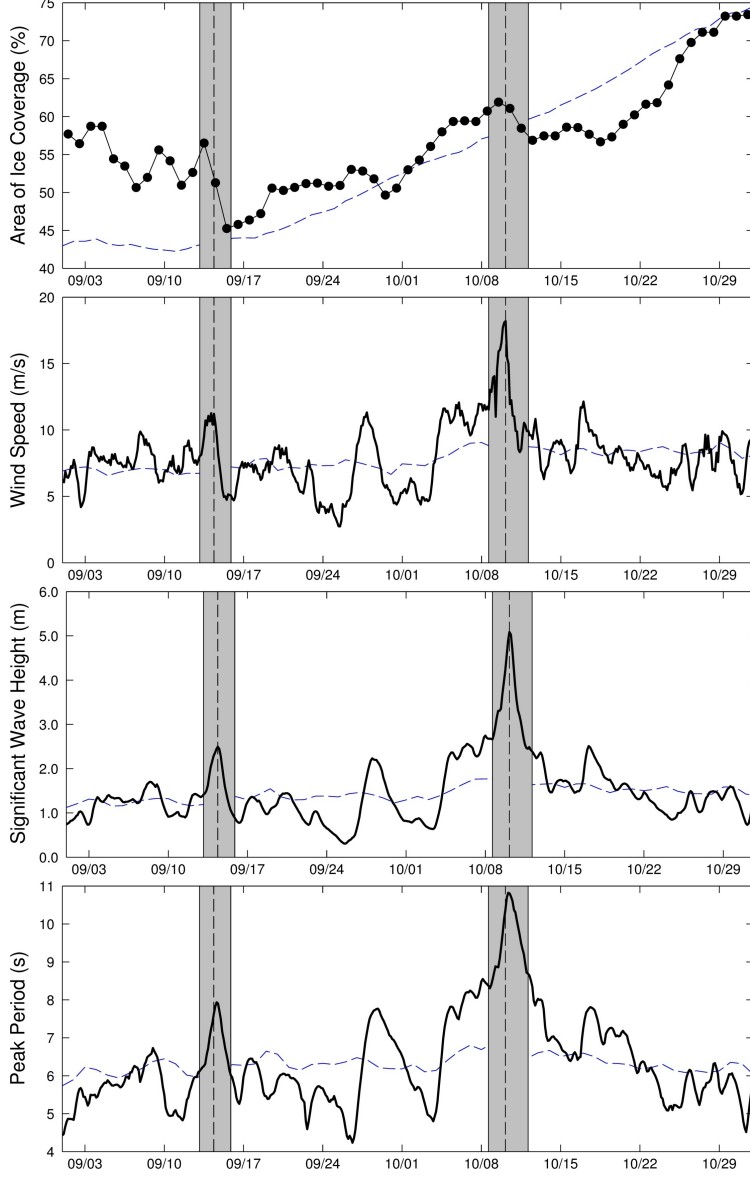

**Figure 12.** Area-averaged time series of ice coverage, wind speed, significant wave height, and peak period in the Beaufort-Chukchi Sea for September-October 2006 (solid line). The dashed line is the daily average from 1992-2014.



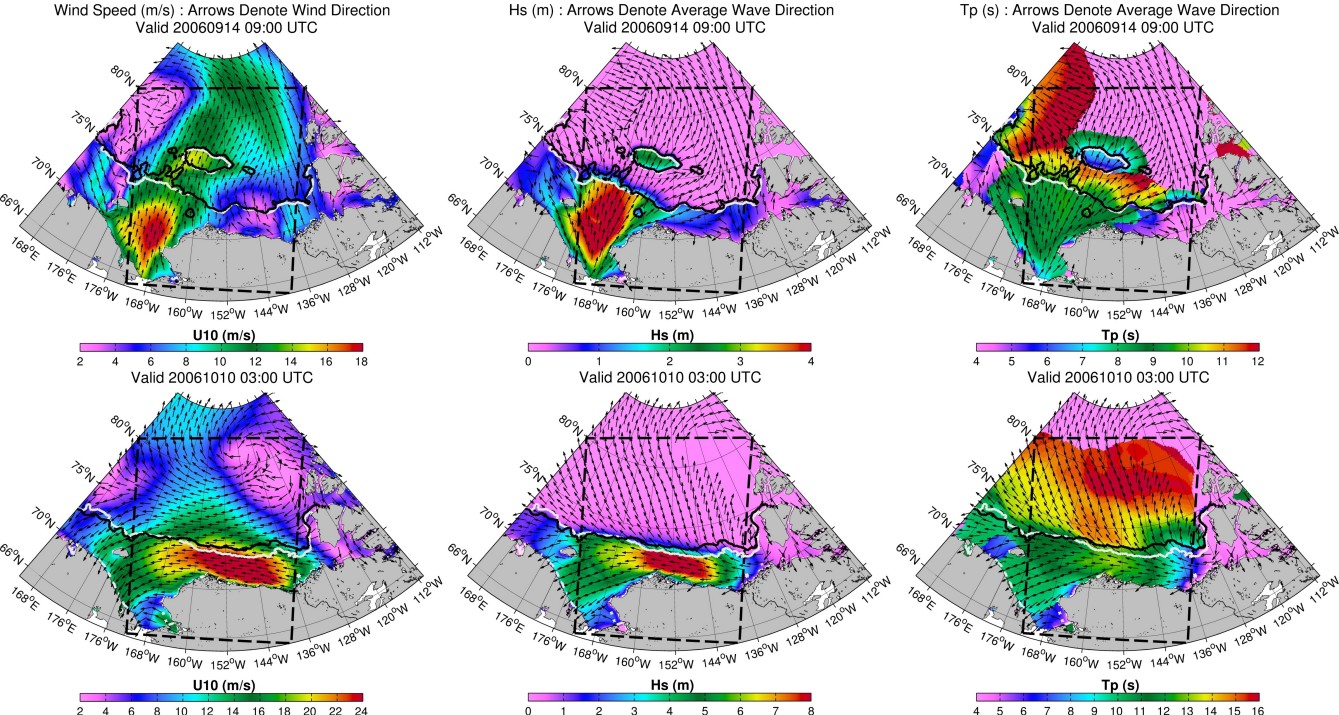

**Figure 13.** Wind speeds, significant wave heights, and peak periods for two selected events in September and October 2006. The arrows denote either the wind direction or average wave direction. The white and black contour lines represent the ice edge defined by a 15% concentration before and after the event.





**Table A1.** $H_s$ error metrics for select years in the Beaufort-Chukchi Sea using CFSR and ERAI in parentheses.

| Buoy ID | Depth (m) | Years Valid (YY) | N | NBIAS (%) | RMSE (m) | SI (%) | R | NSTD (%) |
|---|---|---|---|---|---|---|---|---|
| All | - | 12,13,14 | 7574 | +8.44 (-3.14) | 0.29 (0.25) | 20.71 (20.05) | 0.94 (0.94) | +1.42 (-5.55) |
| WMO48213 | 50.01 | 13 | 1700 | +11.40 (+1.36) | 0.31 (0.28) | 25.99 (26.39) | 0.91 (0.91) | +6.22 (+6.97) |
| WMO48214 | 36.13 | 12,13,14 | 3956 | +7.79 (-3.52) | 0.27 (0.24) | 17.55 (16.43) | 0.94 (0.95) | +0.79 (-9.06) |
| WMO48213 | 41.16 | 12 | 568 | +5.66 (-3.67) | 0.28 (0.23) | 15.68 (13.57) | 0.95 (0.95) | -11.66 (-10.30) |
| WMO48211 | 32.52 | 13 | 1350 | +9.95 (-8.22) | 0.30 (0.26) | 30.72 (27.53) | 0.87 (0.88) | +10.61 (-13.24) |





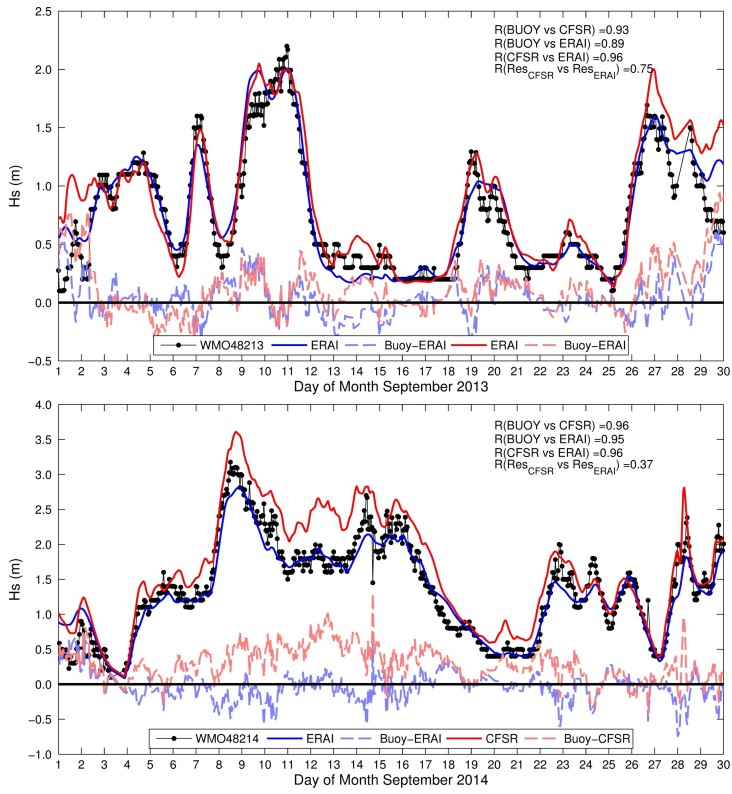

**Figure A1.** Buoy $H_s$ time series for September 2013 (top) and 2014 (bottom). The solid red and blue lines denotes ERAI and CFSR. The dashed lines represent the residual (buoy-model).



**Figure A2.** Significant wave height comparison from CFSR (left) and ERAI (right) versus co-located data from altimeters. The error dispersion of the models is presented in a scatterplot with the density given in a logarithmic scale (top panels). The upper percentiles are highlighted to show the differences (bottom panels).



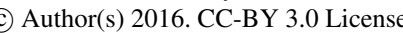



**Figure A3.** The 50th and 99th $H_s$ CFSR percentiles (2010-2014) (top left and middle panels). ERAI-CFSR 50th and 99th percentiles are given in the bottom left and middle panels. The top right panel shows the correlation coefficients between ERAI and CFSR for a monthly averaged time series between CFSR and ERAI. The Mann-Whitney test is presented in the bottom right panel at the 99.9 % confidence limit.





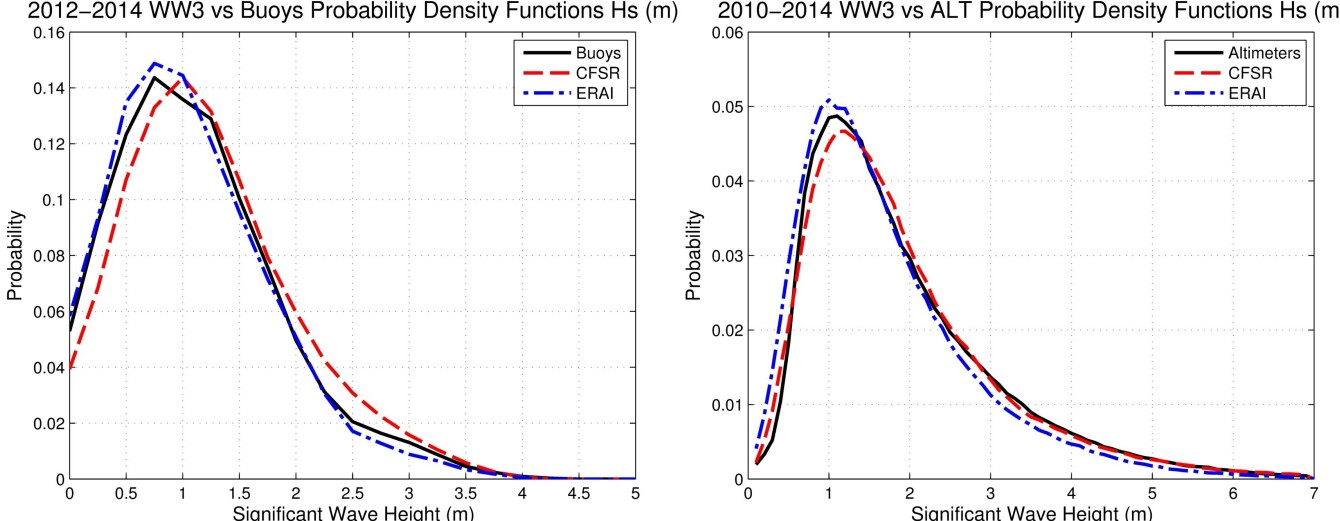

**Figure A4.** $H_s$ probability distributions for CFSR and ERAI versus the buoys (left) and altimeters (right).





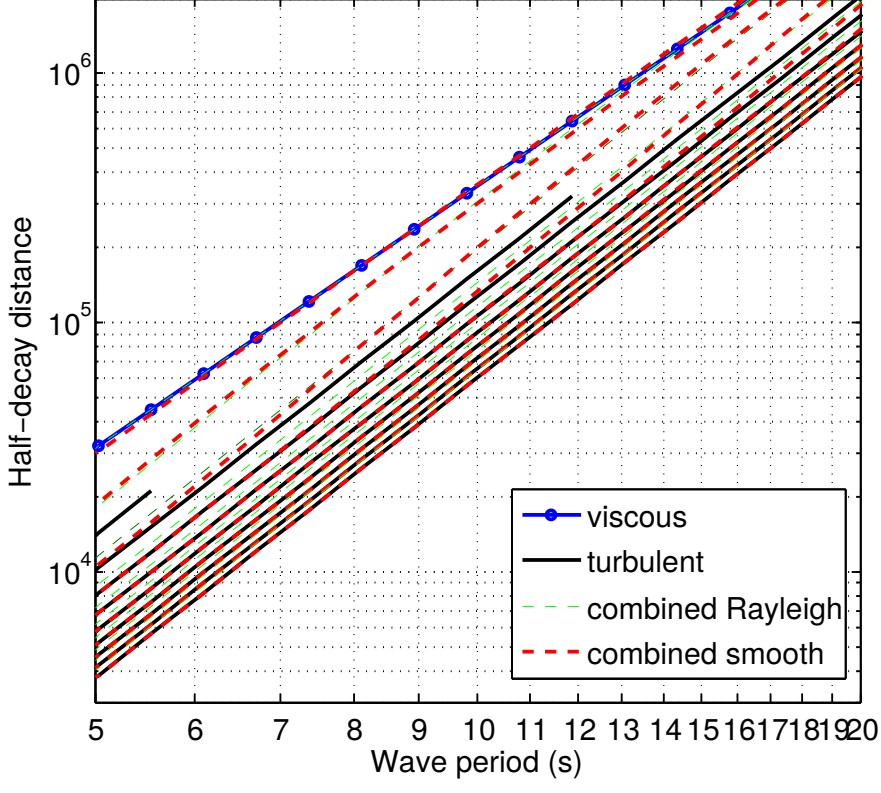

**Figure B1.** Half-decay distances as a function of the wave period $T$, and the significant wave height $H_s$. $H_s$ is varied from 0.5 (upper curves) to 5 m (lower curves). The combination of viscous and turbulent expressions is made using either a Rayleigh distribution of wave height and computing the dissipation for each wave height in the distribution, or by a smooth linear combination of the viscous and turbulent terms adjusted to reproduce the Rayleigh result.





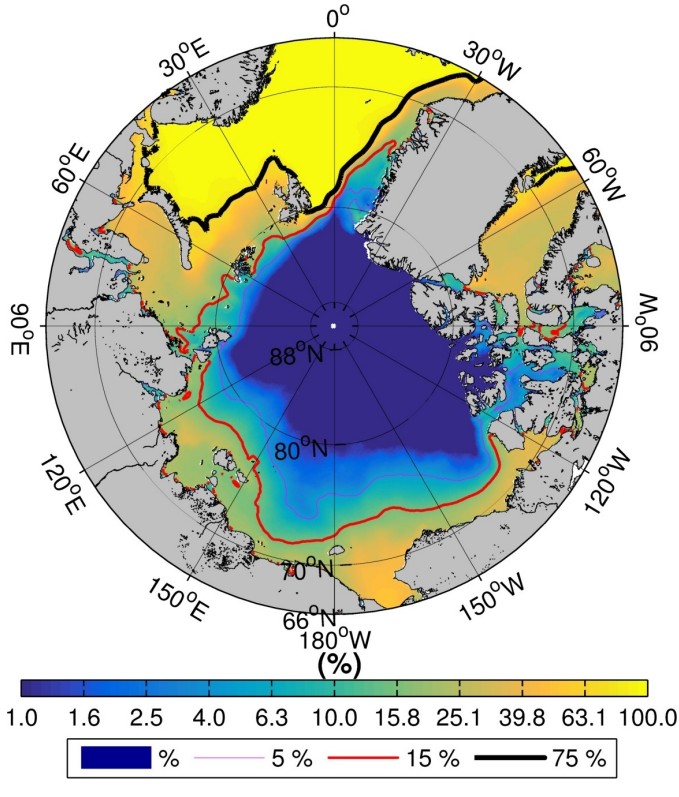

**Figure C1.** Percentage of ice-free time. Contours represent the 5 (thin purple line), 15 (medium red line), and 75 (thick black line).





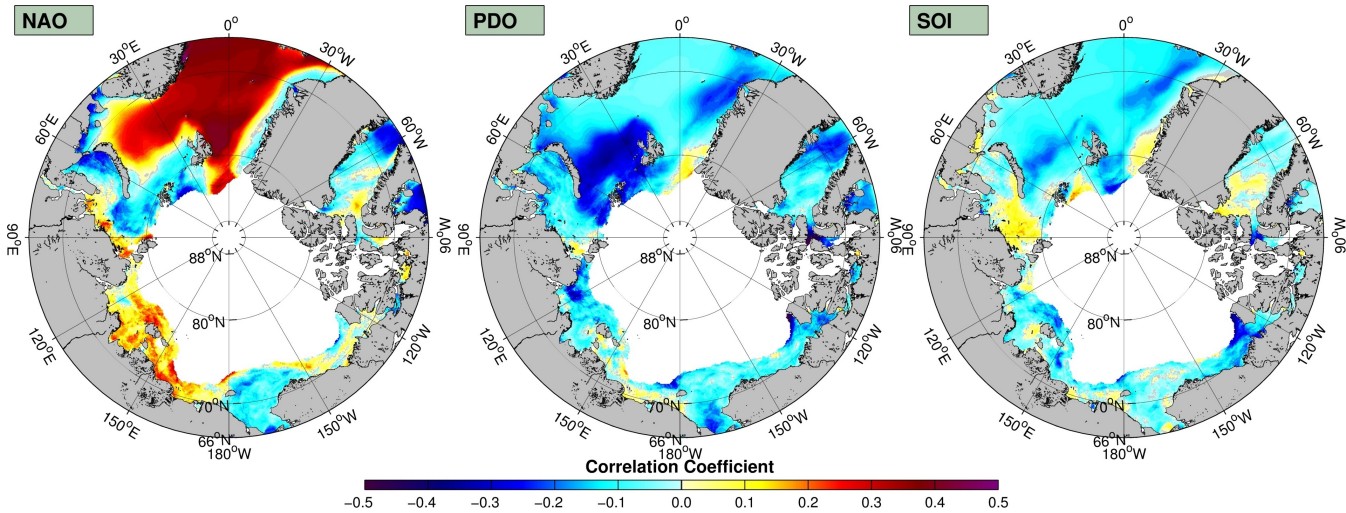

**Figure D1.** Correlation coefficients calculated from monthly time series of $H_s$ of CFSR and the North Atlantic Oscillation (left) and Pacific Decadal Oscillation (right).