# Peer review of "Wave climate in the Arctic 1992-2014: seasonality and trends"

_The Cryosphere, 2016_

## Referee Comment (RC1) · Anonymous Referee #1 · 29 Mar 2016

The paper is a thorough investigation of the changing wave and ice climate in the Arctic ocean. The paper will serve as a baseline study of conditions in the Arctic upon which future studies can build. I recommend the paper is accepted as is.

---

## Referee Comment (RC2) · Anonymous Referee #2 · 24 Apr 2016

Review of Manuscript tc-2016-37: "Wave climate in the Arctic 1992-2014: seasonality and trends", submitted by Stopa et al.

Recommendation: Minor to moderate revisions

The manuscript presents a climatology of the wave climate in the Artic Ocean, for the 1992-2014 period, based on a model hindcast, with buoy and remote sensing altimetry data supporting it. I consider this study as very useful since it addresses a hot topic, regarding the ice cap reduction and the increased open water where waves can be generated. I have some comments and some suggestions that fall into the "minor revisions" category; nevertheless there might be some work ahead. I must admit that I am not satisfied with the writing style. I believe that the first author is an English native speaker, but nevertheless scientific writing has rules and codes that need to

be followed so that the message is properly conveyed and understood. That goal in my opinion is not met, since the style is for most of the times too loose leaving the message hard to understand. The figures captions "suffer" from the same "problem" and are far from complete. For example Figure 1 caption reads: "Regional seas of the Arctic Ocean colors respond to depth and buoy locations are plotted in magenta"; when it might be more complete "Regional seas of the Arctic Ocean, with bathymetry (background contour) and buoy locations (magenta signals)". By the way, consider changing the magenta to another color since it is very hard to see against orange. Also consider adding black lines to the contour, since from 0-500 m is almost impossible to distinguish. The "problem" with Figure 1 caption is transversal to almost all captions, hence the authors should consider revising them where appropriate.

I would like to see the manuscript again.

Specific comments:

P1-L15: "moderates" or "modulates". Either might need revision, since the role of sea ice in the climate system is one of many. P2-L3: replace "manuscript" with "study" (here and hereafter in the text). (It is a manuscript, but it will hopefully be a paper...) P2-L5 (and in the remainder of the paper): you have to be careful with the term "Nordic Seas". Nordic Seas are, to be precise, the passage from the Atlantic to the Arctic Ocean, and comprise the Greenland and the Norwegian Seas. As far as I understand Semedo et al. (2014) has a different geographic span for the Nordic Seas. P2-L11: "can" or "need to"? P2-L18: delete "from" before "derived". P2-L19: sentence starting with "The continuous" is confusing; consider re-writing, please. P2-L20: this sentence needs to be revisited, since it is confusing and not correct. Era-Interim is not a wave hindcast but a wave reanalysis itself, since it is produced with a wave-atmosphere coupled model system (the IFS), with data assimilation. You can find more information about ERA-Interim in Dee et al. (2011) in your reference list. You need to correct the references to Era-Interim as a wave hindcast here and in the remainder of the text. P2-L32: replace "give" with "provide". P3-L1: it is not clear in this section the horizontal resolution

of the final hindcast nor the resolution of the intermediate hindcasts. Where the wind fields interpolated and how? P3-L2: add "oceans" before "basins". P4-L4: "validation and observations"; "confusing re-write, please. Models are not validated but evaluated. Consider re-writing here and in the remainder of the text. Sentence starting with "Satellite" is confusing, consider re-writing. P4-L14: "evolves" or "resolves"? P4-L30: replace "average" with "mean". You have to define the n-th spectral moments or refer to a cited publication (some of the ones in your reference list could do the job well). If mo is the zero-th moment then what is m2? You might want to explain why you are using for mean wave period Tm02 (justify your choice). P5-L31: and "the" and "in" after and before "hindcast", respectively. P6-L6/7: "generalize sea ice"? P6-L8: the Nordic Sea comprises the Greenland and the Norwegian Seas; hence this statement needs to be made clearer; see comment above. P6-L15: what do you mean here with "exposure to the atmosphere"? P6-L17: add "seasons" after "extremes". P6-L20: you have to be more precise and accurate: wind waves are a broad concept that usually comprises all wind generated waves (wind waves = surface gravity waves); hence here you mean wind sea waves, no? I believe you make this type of mistake further in the text, so please check. P6-L21: "waves"? Do you mean wind sea wave heights? Be more precise please. P6-L22: how do you know these waves are generated in the basin? P6-L29: replace "flow" with "propagate". P7-L1: add "mean wave" after "swell" (do the same in the remainder of the text). "storm phases"; what do you mean? P7-L3/4: how do you access the wind sea vs. swell percentages? In which figure(s) can you see this? P7-L9: wht do you mean with "corresponding"? P7-L14: sentence starting with "The wave" is confusing; please re-write. P7-L15: sentence starting with "The directions" is confusing; please re-write. Directions of the median? What do you mean? What are the "directions of the median"? P7-L22: suggest deletion of sentence starting with "The percentiles". Why this statement here? Maybe move it to the discussion section... P7-L25: replace sentence starting with "With the" with: "With the loss pf sea ice wave heights are expected to increase". P8-L2: replace "shows" with "show". P8-L16: "cs" or "cm/s"? P8-L30: why bring in the southern ocean index

here? Correlations of what with what? P9-L11: erase "the" after "decreasing". P9-L13: add reference(s) after processes. P9-L31: replace "creates reduced" with "leads to a reduction of". P10-L16: this is most probably a polar low; please confirm and correct accordingly. P10-L28: erase "the" before "waves". P10-L29: erase "the" before "wave"; replace "model" with "models". P10-L31: "present wave models"? Should it be "as presented (or as parameterized) in present wave models". P10-L32: you are not comparing the hindcast Hs output with the altimeters but with the "altimeter Hs observations", no? P11-L22/29: This paragraph seams, to me, speculative. Or the message the authors want to convey need to be made clearer. Where can we see that "the NAO influences the Nordic-Greenland Sea and it is expected to create the negative trend in U10 and Hs which is unique to the rest of the Arctic that has positive trends". (???) (1) The authors need to add (or replace) to their climate index study the Arctic Oscillation, which is related to the NAO, but plays a much greater role in the higher latitudes. (2) The Nordic-Greenland Sea does not exist; see comment above. (3) The positive and negative phases of the NAO play different roles in the area, and that is never mentioned or dealt with. P12-L7: replace ". And" with "; and", and erase "as important sea ice driver". P12-L9: add some references after "studies". P12-L10: replace "use a" with "used as a". P12-L11: "climate conditions"? What do you mean? How will your study influence or be a basis for "future climate conditions"? Do you mean "future climate studies"? P12-L11: "the seas" or the "sea states"? P13-L3: replace "is with "was"; add "forcings" after "ERAI". P13-L4: replace "buoys and altimeters" with "buoy and altimeter observations". P13-L8: "ground truth"; wave buoys have errors as well. In situ observations are, in principle, the most accurate, but they are not the ground truth". They might be the close we cabn get to that, but they are not that. Please consider re-writing. P12-L23: what do you mean by "same pattern"? P12-L24: The Era-Interim is not a model. P13-L1: what do you mean by "creates a correlation"? P14-L3: what do you mean with "residuals"? P14-L4: I get the impression that you are referring to ERAI and CFSR but you mean your hindcasts forced by Era-Interim and CFSR winds, isn't that so? This should be made clearer because here and in the main body of the

text is a major source of confusion. P17-23: correlations of what with NAO and PDO?

---

## Author Comment (AC1) · 21 May 2016

Thank your for your comments. We revised the article based on your comments. We also re-wrote the majority of the text to improve its clarity. We responded to each of the points raised below in blue font.

Review of Manuscript tc-2016-37: "Wave climate in the Arctic 1992-2014: seasonality and trends", submitted by Stopa et al.

Recommendation: Minor to moderate revisions

The manuscript presents a climatology of the wave climate in the Artic Ocean, for the 1992-2014 period, based on a model hindcast, with buoy and remote sensing altimetry data supporting it. I consider this study as very useful since it addresses a hot topic, regarding the ice cap reduction and the increased open water where waves can be generated. I have some comments and some suggestions that fall into the "minor revisions" category; nevertheless there might be some work ahead. I must admit that I am not satisfied with the writing style. I believe that the first author is an English native speaker, but nevertheless scientific writing has rules and codes that need to be followed so that the message is properly conveyed and understood. That goal in my opinion is not met, since the style is for most of the times too loose leaving the message hard to understand.

The figures captions "suffer" from the same "problem" and are far from complete. For example Figure 1 caption reads: "Regional seas of the Arctic Ocean colors respond to depth and buoy locations are plotted in magenta"; when it might be more complete "Regional seas of the Arctic Ocean, with bathymetry (background contour) and buoy locations (magenta signals)". By the way, consider changing the magenta to another color since it is very hard to see against orange. Also consider adding black lines to the contour, since from 0-500 m is almost impossible to distinguish. The "problem" with Figure 1 caption is transversal to almost all captions, hence the authors should consider revising them where appropriate.
-We modified Figure 1 to have the buoy locations plotted in black, changed the colorbar, added depth contours, and changed the caption to:
"Regional seas of the Arctic Ocean with bathymetry (color), buoy locations (black symbols), and 4000, 2000, 500, and 100 m depth contours (black lines)"

Specific comments:

P1-L15: "moderates" or "modulates". Either might need revision, since the role of sea ice in the climate system is one of many.
-We revised the sentence to simply state that sea-ice plays an important role within the climate:
"Sea ice plays an important role within the climate directly impacting the Earth's albedo, meridional ocean circulation, biologic ecosystems, and human activities."

P2-L3: replace "manuscript" with "study" (here and hereafter in the text). (It is a manuscript, but it will hopefully be a paper...)
-We replaced "manuscript" with study here and throughout the rest of the text.

P2-L5 (and in the remainder of the paper): you have to be careful with the term "Nordic Seas". Nordic Seas are, to be precise, the passage from the Atlatic to the Arctic Ocean, and comprise the Greenland and the Norwegian Seas. As far as I understand Semedo et al. (2014) has a different geographic span for the Nordic Seas.
-Semedo et al.,(2014) defines the Nordic Seas encompassing portions of the North Sea, Norwegian Sea,

and Barents Sea. In general the definitions of these seas are not clear. In references to Semedo et al., (2014) and Reistad et al., (2011) we feel these studies were focused on the Nordic Seas and we did not change the text. In this study our focus is North of 66 degrees so we do not cover the entire Nordic Sea defined by Semedo et al., (2014). Throughout the rest of the article we reference the Nordic-Greenland Sea as encompassing the sea from Greenland to Norway.

P2-L11: "can" or "need to"?
-Good point. We can solely use altimeter data but we will be limited to only analyzing wave heights. The model can estimate various other wave parameters that provide a deeper understanding of the wave climate in the Arctic. We simply change the sentence to:
"...therefore, we use numerical models to provide full space-time details essential for a comprehensive description of the wave conditions."

P2-L18: delete "from" before "derived".
-Done

P2-L19: sentence starting with "The continuous" is confusing; consider re-writing, please.
-This statement no longer appears.

P2-L20: this sentence needs to be revisited, since it is confusing and not correct. Era-Interim is not a wave hindcast but a wave reanalysis itself, since it is produced with a wave-atmosphere coupled model system (the IFS), with data assimilation. You can find more information about ERA- Interim in Dee et al. (2011) in your reference list. You need to correct the references to Era-Interim as a wave hindcast here and in the remainder of the text.
-We properly referenced ERA-Interim as a wave reanalysis and in other places in the text. This information now appears in Section 2. The text is now:
In addition, the wave reanalysis of the European Centre for Medium-Range Weather Forecasts (ECMWF) ERA-Interim (ERAI), which couples the atmosphere and wave model and assimilates altimeter wave data, has improved performance over its predecessor ERA-40 (Dee et al. 2011). ERAI has a spatial resolution of 0.7 degree with wind every 6 hours. The wave reanalysis couples the wave and atmosphere models while assimilating wave data from altimeters (Dee et al. 2011).

P2-L32: replace "give" with "provide".
-Done.

P3-L1: it is not clear in this section the horizontal resolution of the final hindcast nor the resolution of the intermediate hindcasts. Where the wind fields interpolated and how?
-In the introduction we stated that the resolution of our wave model:
"To efficiently hindcast in the Arctic basin we implement WW3 on a curvilinear grid matching the spatial resolution of ice concentrations at 12.5 km (Rogers and Orzech 2013)."
But we revised the manuscript and this information now appears in Section 2:
We implement version 5.08 of WW3, on a curvilinear grid matching the spatial resolution of ice concentrations at 12.5 km. The curvilinear grid is well suited to model waves near the Poles since the geographic distance between nodes is equal making the computation more efficient (Rogers and Orzech 2013). The spectra are composed of 24 directions and 32 frequencies exponentially spaced from 0.037 to 0.7 Hz at a relative increment of 1.1. "

We added the statement in section 2.2 to describe that we used linear interpolation to reformat the forcing wind reanalysis datasets to the computational wave grid:
"The reanalysis winds are linearly interpolated to the wave."

P3-L2: add "oceans" before "basins".
-We replaced basins with oceans.

P4-L4: "validation and observations"; "confusing re-write, please. Models are not validated but evaluated. Consider re-writing here and in the remainder of the text. Sentence starting with "Satellite" is confusing, consider re-writing.
-Good point is it better to use evaluate compared to validate.  However, wave hindcasts can be validated and this documentation is often necessary for future studies that might want to use the wave hindcast. These statements were re-written for clarity:
Altimeter data has provided an ample source of global wave observations and aided in the development and evaluation of spectral wave models  (Ardhuin et al., 2010; Stopa et al., 2016, Chen et al., 2002).

P4-L14: "evolves" or "resolves"?
-We were describing how the spectral wave model accounts for wave energy in space, time and spectral space or "evolves" it.

P4-L30: replace "average" with "mean". You have to define the n-th spectral moments or refer to a cited publication (some of the ones in your reference list could do the job well). If mo is the zero-th moment then what is m2? You might want to explain why you are using for mean wave period Tm02 (justify your choice).
-We replaced average with mean. For completeness we defined the spectral moments in the text.
The wave climate is described using the total significant wave height ($H_{s}$) defined as $H_{s}=4\sqrt{m0}$ where $m0$ is the zeroth moment ($p=0$) of the spectrum ($E(f)$) ($m_{p}=\int_0^\infty (2 \pi f)^{p} E(f) df$), mean wave period ($Tm02=\sqrt{m0/m2}$), and average direction ($\theta_{m}$).
We justified our choice of using Tm02:
The mean wave period has reduced variability compared to other wave period definitions (i.e. peak period or $\sqrt{m-1/m0}$) since it is calculated from the second moment of the wave spectrum.

P5-L31: and "the" and "in" after and before "hindcast", respectively.
-Done.

P6-L6/7: "generalize sea ice"?
-"the" was removed.

P6-L8: the Nordic Sea comprises the Greenland and the Norwegian Seas; hence this statement needs to be made clearer; see comment above.
-This and all other references are made to the Nordic-Greeland Sea.

P6-L15: what do you mean here with "exposure to the atmosphere"?
-The seasonal cycle is anti-symmetric because of the winds are increasing and this overlaps with a

partially ice-free ocean. We modified the sentence to:
"The antisymmetric seasonal cycle is created by the increasing wind speeds coupled with partially ice-free seas in September and October."

P6-L17: add "seasons" after "extremes".
-Done.

P6-L20: you have to be more precise and accurate: wind waves are a broad concept that usually comprises all wind generated waves (wind waves = surface gravity waves); hence here you mean wind sea waves, no? I believe you make this type of mistake further in the text, so please check.
-Good point. Here and througout the rest of the article we use the term "wind-seas" to describe waves with wave ages less than 1.2 following the Pierson and Moskowitz, (1964) definition.

P6-L21: "waves"? Do you mean wind sea wave heights? Be more precise please.
-We were referring to the wind-sea wave heights. The sentence was modified to:
"The resulting wind-sea wave heights exceed 3.5 m while the swell wave heights are smaller and travel from the Southwest."

P6-L22: how do you know these waves are generated in the basin?
-This statement was unclear so we removed it.

P6-L29: replace "flow" with "propagate".
-Done.

P7-L1: add "mean wave" after "swell" (do the same in the remainder of the text). "storm phases"; what do you mean?
-"mean wave" was added after swell and throughout the rest of the text.
-Re: storm phases – We are interpreting the opposite directions in the wind-seas and swells to represent different locations (or times) of storms that traverse the area.. We rewrote the sentence to:
"In the narrow corridor of the Baffin Bay, the wind-sea and swell mean wave directions are opposite and represent different locations of passing storms."

P7-L3/4: how do you access the wind sea vs. swell percentages? In which figure(s) can you see this?
-We are referring the last panel. This panel displays the swell persistence which was defined in Section 2.5. To avoid confusion we rewrote the sentence to specifically reference the bottom right panel:
"The bottom right panel shows the swell persistence is >85\% and exceeds 95\% in the Nordic-Greenland, Barents Seas, and Baffin Bay due to their exposure to the swells generated in the North Atlantic."

P7-L9: what do you mean with "corresponding"?
-In section 2.5 we described how the corresponding directions are calculated for the percentiles. We used the matching index of the Hs percentile to define the mean wave period and averaged wave direction.

P7-L14: sentence starting with "The wave" is confusing; please re-write.
-We rewrote the sentence:
"The corresponding wave directions found by using matching indices of the $H_{s}$ give an indication of the wind-sea and swell events. The median is a mix of numerous wave conditions and thus less representative of distinct wind-sea and swell events."

P7-L15: sentence starting with "The directions" is confusing; please re-write. Directions of the median? What do you mean? What are the "directions of the median"?
-We use the wave direction that matches the Hs percentile. The statement is now:
"The corresponding wave directions found by using matching indices of the Hs give an indication of the wind-sea and swell events."

P7-L22: suggest deletion of sentence starting with "The percentiles". Why this statement here? Maybe move it to the discussion section. . .
-This statement was removed.

P7-L25: replace sentence starting with "With the" with: "With the loss pf sea ice wave heights are expected to increase".
-Done.

P8-L2: replace "shows" with "show".
-Done.

P8-L16: "cs" or "cm/s"?
-The plot is given in % per year so we wanted to give the readers an absolute value in cs.

P8-L30: why bring in the southern ocean index here? Correlations of what with what?
-The El Nino Southern Oscillation (ENSO) and the PDO are related (Frey et al., 2015). We use the SOI as an indication of ENSO and explain why we do not show the SOI results. To avoid confusion we removed the reference of the SOI here. The correlation coefficients are computed from monthly averages and the indices. This statement is in the text:
"Table 1 presents correlation coefficients between area-averaged monthly time series of sea ice, U10, and Hs and the NAO and PDO indices (see Appendix 6 for spatial distribution)."

P9-L11: erase "the" after "decreasing".
-Done.

P9-L13: add reference(s) after processes.
-Several references were added: Squire, (2007), Wadhams and Doble (2009), and Li et al., (2015).

P9-L31: replace "creates reduced" with "leads to a reduction of".
-Done.

P10-L16: this is most probably a polar low; please confirm and correct accordingly.
-We confirmed the event is a low pressure system from the reanalysis dataset and not state this in the text:
"The wind field has a cyclonic pattern confirmed to be a Polar Low centered in..."

P10-L28: erase "the" before "waves".
-Done.

P10-L29: erase "the" before "wave"; replace "model" with "models". P10-L31: "present wave models"? Should it be "as presented (or as parameterized) in present wave models".
-Yes good point. We used your recommendation:

"Furthermore, wind-wave generation in partially ice covered waters is expected to be more complex than as parameterized in present wave models Li et al., (2015)."

P10-L32: you are not comparing the hindcast Hs output with the altimeters but with the "altimeter Hs observations", no?
-We are confused by the question. We compare the Hs from the wave hindcast to the Hs observations from the merged altimeter dataset. In addition, there is further model evaluation in Appendix A using available buoy measurements. The sentence is now:
"Despite these missing physical processes, the 23-year hindcast presented here performs well offshore of the sea ice as demonstrated by the Hs comparison to the altimeter observations."

P11-L22/29: This paragraph seams, to me, speculative. Or the message the authors want to convey need to be made clearer. Where can we see that "the NAO influences the Nordic-Greenland Sea and it is expected to create the negative trend in U10 and Hs which is unique to the rest of the Arctic that has positive trends". (???)
-The correlation coefficients were computed from spatial-monthly averaged wave quantities in Table 1 and the supporting information shows the spatial maps of correlation coefficients between monthly averaged Hs and the NAO. This supports our statement and our interpretations that the NAO is influencing the Atlantic-side of the Arctic. We feel that the NAO is causing the negative trend in the Atlantic-section of the Arctic our arguments follow below. We revised the text:
"In the majority of the Arctic, wave heights are increasing. The only region with decreasing wave heights is in the Nordic-Greenland Sea. In our hindcast time period of 1992-2014, the natural variability of the climate through the NAO and PDO impact the Arctic sea state. The NAO influences the Nordic-Greenland Sea and the negative trends observed in the sea state is expected to be caused by the NAO. The PDO influences the Barents and Kara Seas and the monthly correlation closely aligns with the maximum ice loss. In the Beaufort-Chukchi Sea the PDO plays a minor role in the wind and wave fields, but this should be monitored when the PDO transitions into a positive phase."

(1) The authors need to add (or replace) to their climate index study the Arctic Oscillation, which is related to the NAO, but plays a much greater role in the higher latitudes.
-We choose to use the NAO because the index has a stronger relationship in the wave field than the AO. This is most notable in the Atlantic sector since the NAO is regionally defined index. The same can conclusions can be reached using the NAO or AO so for brevity we chose to use the NAO. The below Figure shows the correlation coefficients computed by the monthly average Hs versus the AO and NAO indices. The spatial patterns are nearly identical.

[Figure]

Correlation coefficients computed between monthly averaged Hs and NAO index (left) and AO (right).

(2) The Nordic-Greenland Sea does not exist; see comment above.
-Throughout the text we use the term Nordic-Greenland Sea to define the sea from Greenland to Norway.

(3) The positive and negative phases of the NAO play different roles in the area, and that is never mentioned or dealt with.
-A positive NAO (or AO) means there is an anomalous low pressure in the Arctic. It is expected that the storm events which are associated with stronger winds and waves occur more often. A negative NAO (or AO) means there is anomalous high pressure in the Arctic. This creates more frequent storm activity further South (40-50 degrees N).
Therefore in the Nordic-Greenland Seas a positive NAO means increased wave activity while a negative phase means reduced sea states.
Since the NAO has been positive 1992-1998 and largely negative from 2006-2012 a negative trend is created throughout the climate study period (1992-2014) (see the Figure below). The correlation coefficients are positive in this region so we conclude that the NAO is cause the negative trend in the average monthly Hs time series that we observed in Figure 7.

[Figure]

NAO index 1992-2014 using a 3 month running mean. The blue line represents the linear trend.

We added text in section 4.3:
"In these seas, a positive NAO phase equates to increased sea states while a negative phases have reduced sea states. The NAO was positive in the beginning of the time period (1992-1998) and negative towards the end of the hindcast (2006-2012). This creates a negative trend and its positive relationship with the sea states and wind suggests the NAO is causing the negative wind and wave trends observed in Nordic-Greenland and Barents Seas"

P12-L7: replace ". And" with "; and", and erase "as important sea ice driver".
-Done.

P12-L9: add some references after "studies".
-We added references to the studies we referenced in the introduction:
"Extending previous studies of Francis et al., (2011),Wang et al., (2015),Semedo et al., (2014) in the Arctic we produced a 23-year wave hindcast from 1992 to 2014 using CFSR winds and ice concentrations from SSM/I."

P12-L10: replace "use a" with "used as a".
-Done.

P12-L11: "climate conditions"? What do you mean? How will your study influence or be a basis for "future climate conditions"? Do you mean "future climate studies"?
-We were stating that the hindcast and the results presented here can be used as a basis for future studies as well as a basis to quantify change especially in future climate scenarios (for example using CIMP5 projections). We rewrote the sentence:
"As the Arctic continues to evolve the results presented here can be used as a basis for future climate studies or projections such as those presented by Khon et al.,(2014) or Dobrynin et al., (2012)."

P12-L11: "the seas" or the "sea states"?
-Done.

P13-L3: replace "is with "was"; add "forcings" after "ERAI".
-Done.

P13-L4: replace "buoys and altimeters" with "buoy and altimeter observations".
-Done. The statement is now:
"Measured wave data is essential for validation and we use buoy and altimeter observations."

P13-L8: "ground truth"; wave buoys have errors as well. In situ observations are, in principle, the most accurate, but they are not the ground truth". They might be the close we cabn get to that, but they are not that. Please consider re-writing.
-That is true.  The sentence no longer appears.

P12-L23: what do you mean by "same pattern"?
-This statement was rewritten:
"The scatter indices and correlation coefficients for the wave ERAI and CFSR hindcasts are similar at each buoy."

P12-L24: The Era-Interim is not a model.
-We rewrote the majority of the references in this section to ERAI hindcast to avoid confusion with the ERA-Interim reanalysis.

P13-L1: what do you mean by "creates a correlation"?
-In general the CFSR hindcast has more variability and this causes lower correlation coefficients.  The ERAI hindcast is much smoother and in general the correlations coefficients are better.  This statement was rewritten:
"The ERAI hindcast's time series is smoother and has a correlation coefficient of 0.89."

P14-L3: what do you mean with "residuals"?
-In the Figure we show the residuals and show the correlation coefficient of the residuals (Hindcast-Buoy) (i.e the correlation of the CFSR hindcast-buoy vs  the ERAI hindcast-buoy).   We re wrote the sentence to:
"The hindcast residuals (CFSR-buoy and ERAI-buoy hindcasts) are moderately correlated with coefficients of 0.75 showing the forcing wind fields are similar."

P14-L4: I get the impression that you are referring to ERAI and CFSR but you mean your hindcasts forced by Era-Interim and CFSR winds, isn't that so? This should be made clearer because here and in the main body of the text is a major source of confusion.
-Yes we were using CFSR and ERAI to describe these two 5 year wave hindcasts.  We re-wrote this entire section to referring to them as the CFSR and ERAI hindcasts.

P17-23: correlations of what with NAO and PDO?
-We rewrote the statement to specifcally describe the correlation coefficients:
"This study presents area-average correlation coefficients computed between monthly Hs and the NAO and PDO indices to quantify the strength of the relationship."